# Physiologically relevant reconstitution of iron-sulfur cluster biosynthesis uncovers persulfide-processing functions of ferredoxin-2 and frataxin

Sylvain Gervason[1,7], Djabir Larkem [1,7], Amir Ben Mansour[1,7], Thomas Botzanowski[2], Christina S. Müller[3], Ludovic Pecqueur [4], Gwenaelle Le Pavec[1], Agnès Delaunay-Moisan[1], Omar Brun[5], Jordi Agramunt [5], Anna Grandas [5], Marc Fontecave [4], Volker Schünemann[3], Sarah Cianférani[2], Christina Sizun [6], Michel B. Tolédano[1] & Benoit D'Autréaux[1]

Iron-sulfur (Fe-S) clusters are essential protein cofactors whose biosynthetic defects lead to severe diseases among which is Friedreich's ataxia caused by impaired expression of frataxin (FXN). Fe-S clusters are biosynthesized on the scaffold protein ISCU, with cysteine desulfurase NFS1 providing sulfur as persulfide and ferredoxin FDX2 supplying electrons, in a process stimulated by FXN but not clearly understood. Here, we report the breakdown of this process, made possible by removing a zinc ion in ISCU that hinders iron insertion and promotes non-physiological Fe-S cluster synthesis from free sulfide in vitro. By binding zinc-free ISCU, iron drives persulfide uptake from NFS1 and allows persulfide reduction into sulfide by FDX2, thereby coordinating sulfide production with its availability to generate Fe-S clusters. FXN stimulates the whole process by accelerating persulfide transfer. We propose that this reconstitution recapitulates physiological conditions which provides a model for Fe-S cluster biosynthesis, clarifies the roles of FDX2 and FXN and may help develop Friedreich's ataxia therapies.

[1] Institute for Integrative Biology of the Cell (I2BC), CEA, CNRS, Univ. Paris-Sud, Université Paris-Saclay, 91198 Gif-sur-Yvette cedex, France. [2] Laboratoire de Spectrométrie de Masse BioOrganique, Université de Strasbourg, CNRS, IPHC UMR 7178, 67000 Strasbourg, France. [3] Fachbreich Physik, Technische Universität Kaiserslautern, Erwin-Schrödinger-Str. 56, 67663 Kaiserslautern, Germany. [4] Laboratoire de Chimie des Processus Biologiques, Collège de France, Sorbonne Université, CNRS UMR 8229, PSL Research University, 11 place Marcelin Berthelot, 75005 Paris, France. [5] Departament de Química Orgànica i IBUB, Facultat de Química, Universitat de Barcelona, Martí i Franquès 1-11, E-08028 Barcelona, Spain. [6] Institut de Chimie des Substances Naturelles, CNRS, Université Paris Saclay, 91190 Gif-sur-Yvette, France. [7] These authors contributed equally: Sylvain Gervason, Djabir Larkem, Amir Ben Mansour. Correspondence and requests for materials should be addressed to B.D. (email: benoit.dautreaux@i2bc.paris-saclay.fr)

ron-sulfur (Fe-S) clusters are highly conserved and ubiquitous prosthetic groups of proteins, made of iron and sulfide ($S^{2-}$) ions, of which the [2Fe2S] and [4Fe4S] clusters are the most common in biology[1–4]. Organisms from the three life kingdoms: archea, bacteria, and eukaryota have exploited the versatile properties of Fe-S clusters to perform essential biological functions, including ATP production, Krebs cycle, protein synthesis, and maintenance of genome integrity[5,6]. The defects in Fe-S cluster biosynthesis lead to severe human pathologies, which underscores the importance to unravel the mechanism of their assembly[2,7–12]. Fe-S clusters are biosynthesized de novo by specialized multi-protein machineries via a process conserved from bacteria to eukaryotes[1,3,4,6,13]. In mitochondria, Fe-S clusters are assembled by the Iron Sulfur Cluster assembly machinery (ISC), which encompasses the scaffold protein ISCU, the NFS1-ISD11-ACP complex containing the cysteine desulfurase NFS1, a pyridoxal-phosphate (PLP) enzyme that generates a precursor of sulfide in the form of a cysteine-bound persulfide (Cys-SSH) by desulfurization of L-cysteine, ferredoxin 2 (FDX2) and its cognate reductase (FDXR) which together deliver electrons provided by NADPH and frataxin (FXN) which is required for efficient Fe-S cluster biogenesis[1–4,9]. However, how the ISC machinery operates is not clearly understood and thereby, the biochemical roles of FDX2 and FXN are not established[1,3,9]. Fe-S clusters biosynthesis is thought to rely on confined production of sulfide in the proximity of iron to promote formation of the iron-sulfide bond while preventing toxic diffusion of iron and sulfide ions, but no data have documented such a confined synthesis[1,3].

Several reconstitutions of both, bacterial and eukaryotic ISC machineries have attempted to address these questions[14–21]. Bacterial ISC reconstitutions indicate that the homologue of NFS1, IscS, transfers several persulfides to multiple cysteines on the ISCU homologue, IscU, which suggests that sulfide production takes place on the scaffold protein[22,23]. In mammals too, persulfides are transferred to ISCU, but only one of its cysteines becomes persulfidated[14,24]. These reconstitutions have, however, not answered the question of the physiological reductant of the persulfide and whether its reduction is coupled to the presence of iron. The FDX2-FDXR reducing system is a good candidate for this reaction, but most reconstitutions used the non-physiological reductant dithiothreitol (DTT) instead, thereby occulting the possible function of FDX2 in this process[15,16,18–20]. Moreover, DTT-dependent Fe-S cluster reconstitution cannot be considered as reproducing a physiological process since DTT reduces the persulfide of the cysteine desulfurase, which leads to formation of free sulfide that is not confined to ISCU[14,21]. DTT was also shown to promote formation of [4Fe4S] clusters, the biosynthesis of which is apparently supported by the ISCA type proteins[25–27]. The first study of the role of FDX2 was performed with bacterial Fdx2 and indicated that it promotes reductive coupling of [2Fe2S] clusters into [4Fe4S], which may not be physiologically relevant as formation of [4Fe4S] clusters on ISCU is possibly non-natural[17]. A FDX2-dependent assembly of [2Fe2S] clusters on ISCU was then reported with yeast proteins, thus suggesting that FDX2 is an important component, but evidence that FDX2 reduces the persulfide was not provided[21]. Moreover, we found that FDX2 is not able to reduce the persulfide transferred to ISCU, which questioned its role in persulfide reduction[14].

Similarly, the role of frataxin (FXN) remains very controversial despite tremendous efforts to understand its biochemical function[9]. Since the discovery that Friedreich's ataxia (FA), a severe neurodegenerative and cardiac disease that is the most common form of recessive ataxia is caused by defective expression of FXN, this protein has been the focus of intense research[2,9,10,28]. FXN was first proposed to operate as an iron chaperone or iron storage protein providing iron to the ISC machinery, but this model was challenged by several in vivo studies[9,29–31]. The reconstitutions of the ISC machinery with FXN did not provide a clearer picture either[14,15,18,20,32]. The first reconstitution was performed with the bacterial system and unexpectedly showed that the bacterial homolog of FXN, CyaY, inhibits the rate of Fe-S cluster assembly under iron rich conditions, by slowing down sulfide release by IscS, which was interpreted as a mechanism needed to prevent uncontrolled Fe-S cluster formation[15,32]. The reconstitutions of the mammalian system showed in contrast that FXN stimulates sulfide production and concomitantly iron entry in the ISC complex[14,18,20]. However, these studies were performed with DTT instead of the FDX2-FDXR reducing system, which again questions their physiological relevance. Another study reported that yeast FXN stimulates persulfide formation on NFS1[33], but this could not be reproduced with the murine proteins[14]. The only reconstitution including both FDX2 and FXN concluded that FXN is strictly required for Fe-S cluster assembly, which may not be consistent with the dispensable role of FXN in vivo[21,31,34–36]. FXN was also reported to stimulate persulfide transfer to ISCU, therefore promoting confined production of sulfide[14,24]. However, FDX2 was unable to reduce the persulfide on ISCU, which prevented a direct correlation with Fe-S cluster assembly[14].

Several studies also attempted to assess the iron binding properties of ISCU[19,37–39]. Bacterial IscU was shown to bind iron in the μM range but another study could not detect interaction with either $Fe^{2+}$ or $Fe^{3+}$ ions[19,39]. Yeast and drosophila ISCU proteins were reported to bind iron but not in the assembly site and it was not shown whether this iron containing form could sustain Fe-S cluster assembly[37,38].

We report here that a clue to these discrepancies is the presence of a zinc ion in the assembly site of ISCU, which has been persistently reported in bacterial and eukaryotic ISCU proteins[40–45]. We show that this zinc ion hinders iron binding and precludes the reduction of the persulfide of ISCU by FDX2, thereby fostering reduction of the persulfide of NFS1 by L-cysteine, which leads to release of free sulfide and Fe-S cluster formation that cannot be considered as physiologically relevant since it is not confined to ISCU. By exchanging zinc with iron, we generate an iron-loaded ISCU protein allowing Fe-S clusters synthesis via FDX2-dependent reduction of the persulfide of ISCU. In this process, FXN is not required for iron insertion but stimulates persulfide transfer to ISCU. Moreover, both persulfide transfer and reduction require iron, which most likely ensures the coordination between sulfide production and iron availability in ISCU. We propose that the reaction performed by iron-loaded ISCU reproduces the physiological process of Fe-S cluster assembly, thus allowing the elucidation of the sequence of Fe-S cluster biosynthesis and the respective roles of FXN and FDX2.

## Results

**ISCU binds a $Fe^{2+}$ ion upon removal of the $Zn^{2+}$ ion.** To address these questions, we first attempted to isolate an iron containing form of ISCU that would be competent for Fe-S cluster assembly. Analysis of the metal content of purified mouse ISCU revealed the presence of zinc as previously reported, but no iron[40–42,46]. We measured $0.8 \pm 0.1$ zinc ion/ISCU and up to $1.0 \pm 0.1$ by incubation with additional zinc (Fig. 1a). Site-directed mutagenesis confirmed that the conserved amino acids of the Fe-S cluster assembly site of ISCU, C35, D37, C61, and H103, but not C104, are required for zinc binding, as also observed in the structures of human ISCU (Fig. 1a)[43]. To probe binding of iron to the assembly site of ISCU, we sought for typical ligand-to-metal charge transfer (LMCT) bands of ferrous iron bound to cysteine, expected in the near UV domain[47,48]. No such feature appeared

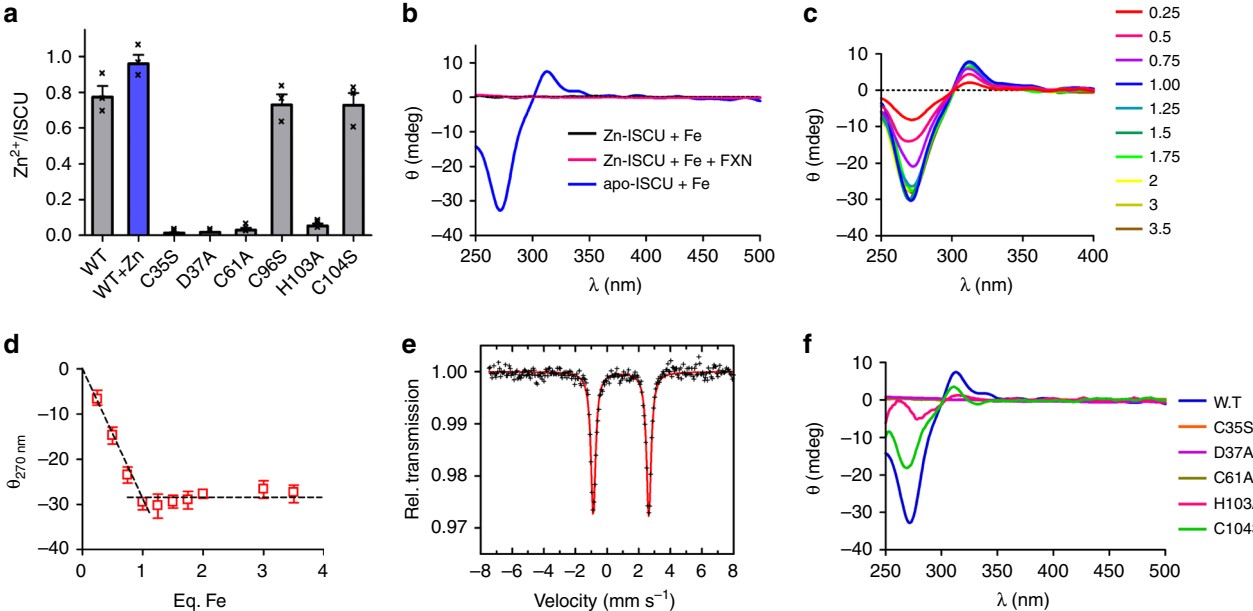

**Fig. 1** Characterization of iron-loaded ISCU. **a** Zinc content of WT ISCU as purified (gray) and after zinc repletion (blue), and of the C35S, D37A, C61A, C96S, H103A, and C104S mutants, indicated as molar ratio of zinc ion per ISCU protein. **b** CD spectra of Zn-ISCU with (pink) and without FXN (black) and of apo-ISCU (blue), incubated with one molar equivalent of $Fe^{2+}$ ions. **c** Titration of apo-ISCU with $Fe^{2+}$ ions monitored by CD. Amounts of iron are indicated as molar ratio of iron to ISCU. **d** Plot of the values of the CD signal at 270 nm from the spectra in **c** at each concentration of iron. **e** Mössbauer spectrum of apo-ISCU incubated with one equivalent of $^{57}Fe$ enriched $Fe^{2+}$ ions. The red solid line represents the best fits of the data achieved using the components displayed as black solid lines. See Supplementary Table 1 for parameters. **f** CD spectra of apoISCU proteins WT (blue), C35S (orange), C61A (brown), D37A (purple), H103A (pink), and C104S (green) incubated with one molar equivalent of $Fe^{2+}$ ions. Errors bars represent standard deviation from $n = 3$ independent experiments and source data for panels **a**, **d** are provided as a Source Data file

upon addition of $Fe^{2+}$ to the zinc containing form of ISCU (Zn-ISCU), alone or in the presence of FXN (Fig. 1b). In contrast, adding $Fe^{2+}$ to ISCU devoid of metal (apo-ISCU) produced absorptions at 270 nm, 310 nm, and 340 nm that are characteristic of Cys-S → Fe(II) LMCT bands in the circular dichroism (CD) spectrum (Fig. 1b). Iron titrations by CD showed that ISCU binds ~0.95 $Fe^{2+}$ ions in this site (Fig. 1c, d). Altogether, these data also show that FXN is not able to exchange zinc for iron and is neither required for iron insertion in apo-ISCU (Fig. 1b). Moreover, we did not detect any iron or zinc in purified FXN, which strengthened the idea that FXN is not involved in iron insertion.

We next evaluated the effect of iron on the structure of apo-ISCU by NMR spectroscopy, since ISCU was reported to exist in two different conformational states, structured (denoted S) and disordered (denoted D), and zinc was shown to stabilize the S state[49]. The NMR spectrum of apo-ISCU displayed broad central signals that were assigned to the D state and more dispersed resonances corresponding to the S state (Supplementary Fig. 1a). Integration of the tryptophan W74 signals indicated that ~30% of apo-ISCU was in the S form (inset of Supplementary Fig. 1a). Upon addition of iron, the central part of the spectrum became more resolved and additional dispersed signals appeared (Supplementary Fig. 1b). Two signals were visible for W74: an intense one slightly shifted relative to that of the S form of apo-ISCU, and a weak one at the position of the D form of apo-ISCU. By analogy, we assigned the weak signal to remaining apo-ISCU in the D form and the shifted intense signal to an iron containing S form. Integration of the W74 signals indicated that ~80% of ISCU was in the S state.

To further assess the iron binding properties of ISCU, we used Mössbauer spectroscopy. The Mössbauer spectrum of apo-ISCU incubated with one equivalent of $Fe^{2+}$ revealed the presence of two species (Fig. 1e, Supplementary Table 1). The parameters of

the major component (component 1, 85%) were indicative of a high spin Fe(II) center coordinated in a mixed environment by one or two cysteines and two or three N/O ligands, consistent with the CD spectrum (Supplementary Tables 1 and 2 and Supplementary Note 1)[50]. The high value of the quadrupole splitting (3.51 mm.s$^{-1}$), that is growing with asymmetry, is in agreement with this mixed coordination. Remarkably, the amount of component 1 was similar to the proportion of ISCU in the S state, which suggests that binding of iron to this site promotes conversion of the D to S state. In contrast, the isomeric shift of the minor species (component 2, 15%) was consistent with a Fe(II) center coordinated by five or six N/O atoms but no sulfur (Supplementary Tables 1 and 2). The parameters of this species were not identical to those of free iron and this species was absent when apo-ISCU was incubated with a sub-stoichiometric amount of iron, which suggests that component 2 is iron bound to ISCU in a lower affinity site compared to component 1 (Supplementary Fig. 1c, d and Supplementary Table 1).

To identify the ligands of the iron ion in these sites, we used site-directed mutagenesis. The C35S, D37A, and C61A substitution mutants lacked the typical LMCT bands when incubated with ferrous iron (Fig. 1f). Very weak absorptions were detected for the H103A mutant but its Mössbauer spectrum lacks the typical signal of the iron-cysteinyl site (Fig. 1f, Supplementary Fig. 1e and Supplementary Note 1). In contrast, the C104S mutant exhibited intense Cys-S → Fe(II) LMCT absorption bands and the major species detected by Mössbauer spectroscopy (65%) displayed parameters nearly identical to those of the WT protein (Fig. 1f, Supplementary Fig. 1f and Supplementary Note 1). The lack of the minor species (component 2) in the H103A and C104S mutants further suggested that H103 is a ligand of this iron ion and that the C104S mutation impairs iron binding to this site (Supplementary Note 1). We conclude that the zinc ion prevents iron binding in the assembly site of ISCU, which may explain the

lack of interaction with iron previously observed[38,39]. Removing zinc allows binding of a $Fe^{2+}$ ion in the assembly site via the C35, D37, C61, and H103 amino acid residues, which stabilizes the S state of ISCU. The minor component may correspond to iron bound in a distorted assembly site via H103 in the D state.

**Physiologically relevant assembly of Fe-S cluster by Fe-ISCU.** We then tested the ability of both, Fe-ISCU (apo-ISCU incubated with one equivalent of $Fe^{2+}$ ions) and Zn-ISCU in the presence of one equivalent of iron, to assemble a Fe-S cluster, under catalytic conditions with all the components of the ISC machinery (the NFS1-ISD11-ACP (NIA) complex, FXN, FDX2 and FDXR) at a 1:10 molar ratio relative to ISCU, NADPH as a source of electrons and L-cysteine in stoichiometric amounts relative to ISCU. Fe-S cluster assembly was monitored by CD and UV-visible spectroscopies. The reaction with Fe-ISCU led to appearance of a species within 3 min, with spectroscopic features identical to those reported for the oxidized form of the [2Fe2S] cluster in ISCU (Fig. 2a, b)[16,21]. In contrast, the reaction with Zn-ISCU generated virtually no Fe-S cluster (Fig. 2a), in agreement with the hindrance of iron binding in the assembly site when zinc is bound (Fig. 1b). Mössbauer spectroscopy and native mass spectrometry showed that exclusively $[2Fe2S]^{2+}$ clusters were generated, in contrast to previous reconstitutions which reported formation of both [2Fe2S] and [4Fe4S] clusters (Fig. 2c, d, Supplementary Fig. 1h, i and Supplementary Note 2)[16,18].

Moreover, native mass spectrometry showed that the [2Fe2S] cluster was hosted in a monomer of ISCU (Fig. 2d). The identification of two different iron binding sites by Mössbauer spectroscopy further indicated that the [2Fe2S] cluster was in an asymmetrical arrangement, in agreement with crystal structures and spectroscopic studies suggesting a three Cys, one Asp mixed coordination in a monomer of ISCU (Fig. 2c, Supplementary Note 2)[17,45,51,52].

Iron and cysteine titrations showed that approximately two irons and two cysteines per ISCU were required to form the [2Fe2S] cluster when starting from apo-ISCU, with yields in the range of 90% of reconstituted ISCU, which confirms the stoichiometry of one [2Fe2S] cluster per ISCU (Fig. 2e, f). Importantly, these data also indicate that the reaction is highly efficient, with processing of nearly all L-cysteine and iron added into [2Fe2S] clusters. Similar spectra were collected when the reaction was performed with stoichiometric amounts of the NIA complex, ISCU and FXN, which attested that the Fe-S cluster is generated within the NFS1-ISD11-ACP-ISCU-FXN (NIAUF) complex (Supplementary Fig. 2a).

We then evaluated the dependence of this reaction on FXN and FDX2. In reactions missing FDX2, the assembly was compromised, consistent with the absolute cellular requirement of FDX2 for Fe-S cluster biogenesis (Fig. 2g)[53,54]. When FXN was omitted, the assembly was slowed down, consistent with the significant decrease of Fe-S cluster biogenesis activity in cells lacking FXN (Fig. 2g)[31,34,35]. In conclusion, the reconstituted system with

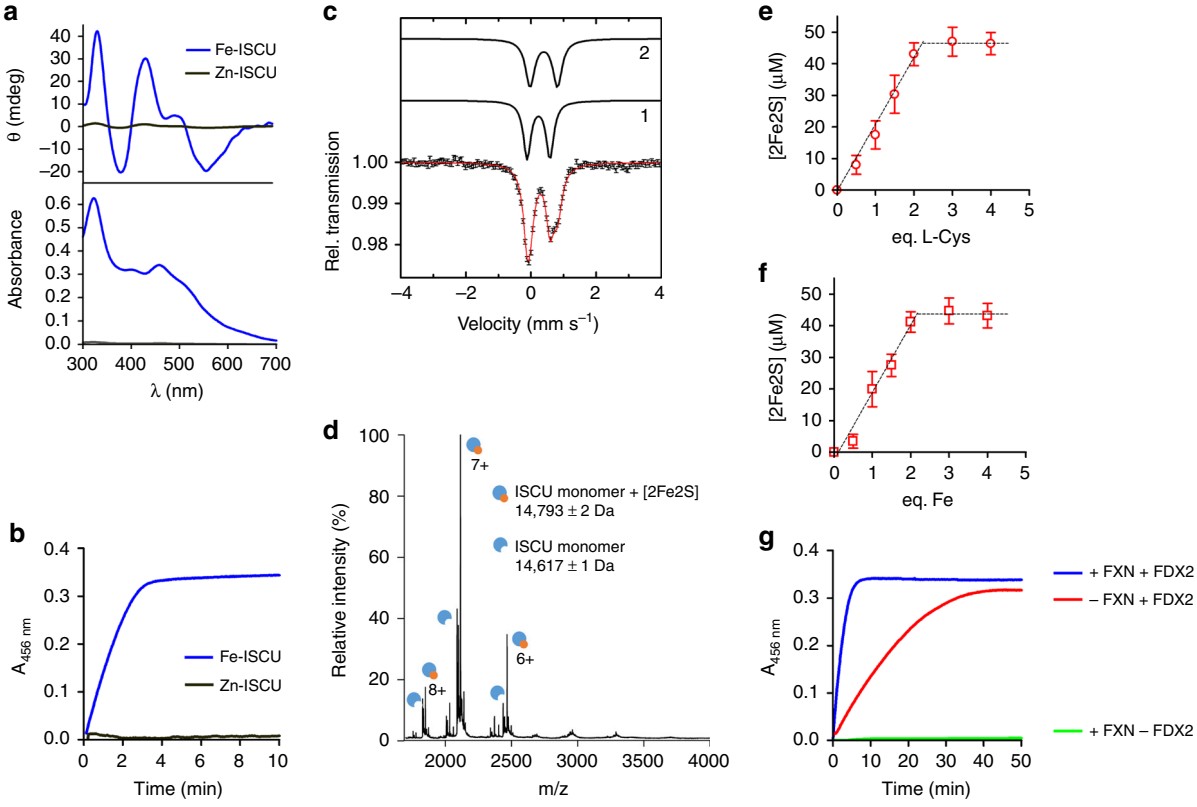

**Fig. 2** Fe-S cluster assembly by Fe-ISCU and Zn-ISCU. **a** UV and CD spectra of Fe-S cluster reconstitution reactions by Fe-ISCU (blue) and Zn-ISCU (black) performed under catalytic conditions with one equivalent of L-cysteine. One equivalent of iron was added in the case of Zn-ISCU. **b** Kinetics of Fe-S cluster reconstitution by Fe-ISCU (blue) and Zn-ISCU (black) as described in **a**, monitored at 456 nm. **c** Mössbauer spectrum of reconstitution performed with apo-ISCU in the presence of two equivalents or iron and L-cysteine. The red solid line represents the best fit of the data achieved using the two components displayed as black solid lines (See Supplementary Table 1 for parameters). **d** Native MS spectra of reconstitution performed with apo-ISCU in the presence of two equivalents or iron and L-cysteine. **e**, **f** Titrations of [2Fe2S] clusters in ISCU for various amounts of L-cysteine (**e**) and iron (**f**) (Source data are provided in Supplementary Fig. 2b, c and as a Source Data file). **g** Fe-S cluster reconstitution assays by Fe-ISCU in reactions containing all components as described in **a** (blue) and missing FDX2 (green) or FXN (red)

Fe-ISCU appears to recapitulate biological conditions of Fe-S cluster assembly. Furthermore, the reaction performed without FXN led to formation of a [2Fe2S] cluster with identical features as those observed in the presence of FXN and in similar yields as the complete reaction, thus suggesting that FXN does not change the outcome of the reaction, but only modulates its rate (Fig. 2g, Supplementary Fig. 2d).

**Mechanism of FDX2-based Fe-S cluster assembly by Fe-ISCU.** We next examined how the persulfide of NFS1 was reduced and sulfide incorporated as a Fe-S cluster. We previously showed using a protein persulfidation assay that NFS1 transfers its per-sulfide to the cysteine C104 of ISCU[14]. Using this assay, we monitored persulfide transfer to Fe-ISCU and Zn-ISCU and found that both acquire a persulfide from NFS1, on a single cysteine (Fig. 3a). Lack of transfer with the C104S mutant provided evidence that this residue serves as the persulfide receptor, and mass spectrometry indicated that a single sulfur atom is incorporated in ISCU (Fig. 3b–d)[14]. In the absence of metal, persulfide transfer was abolished (Fig. 3b), despite formation of a complex between apo-ISCU, FXN, and the NIA complex (Supplementary Fig. 3a and Supplementary Note 3), which indicates that the metal ion is required for persulfide transfer. Adding FDX2 to the persulfidated form of Fe-ISCU led to disappearance of the persulfide and concomitant formation of a Fe-S cluster, thus indicating that FDX2 reduces the persulfide into sulfide (Fig. 3a **left panel, 3e, 3f**). In contrast, FDX2 was unable to reduce the persulfide of Zn-ISCU, which explains the inability of this form to assemble a Fe-S cluster (Fig. 3a right panel). Per-sulfide reduction was also precluded in the absence of metal, which rules out the idea that zinc could be inhibitory and instead points to a specific requirement of iron for persulfide reduction (Fig. 3g).

Since formation of a [2Fe2S] cluster requires two sulfides, we asked whether a second sulfide was provided by NFS1 either directly, by FDX2-mediated reduction of the persulfide of NFS1, or indirectly, via a second transfer-reduction process following reduction of the first one. In both cases, a concomitant disappearance of the persulfides of NFS1 and ISCU was expected upon addition of FDX2. To prevent reloading of NFS1 with a persulfide, which would otherwise mask both direct reduction and second transfer, we used sub-stoichiometric amounts of L-cysteine. These experiments were performed by incubating reaction mixtures containing the NFS1-Fe-ISCU complex with various amounts of L-cysteine from sub-stoichiometric to excess, prior to adding FDX2. The persulfidation states of NFS1 and ISCU were analyzed before and after addition of FDX2. Upon addition of FDX2, the persulfide of ISCU was reduced as described in the previous section, but the persulfide of NFS1 did not disappear, thus establishing that only one sulfide is provided to ISCU (Fig. 3h, i). As ISCU binds only one iron ion, formation of a [2Fe2S] cluster likely requires dimerization of ISCU followed by segregation of the Fe-S cluster on one of the two subunits.

**FXN stimulates persulfide transfer to Fe-ISCU.** We next inspected the effect of FXN on each step of the assembly process. The yeast FXN homologue was proposed to stimulate persulfide formation on NFS1[33]. However, this effect could not be observed with the murine system, therefore indicating that FXN operates at a later step[14]. We first tested the effect of FXN on persulfide transfer. We previously reported that persulfide formation on NFS1 is much faster than persulfide transfer, i.e., that persulfide transfer is the rate-limiting step of this process[14]. Thereby, the global rate of persulfidation of ISCU upon addition of L-cysteine to the NIAU

complex, which encompasses both, persulfide formation on NFS1 and its transfer to ISCU, only reflects persulfide transfer, thus allowing direct determination of the rate of persulfide transfer by measuring the rate of the two-step reaction. In the absence of FXN, persulfide transfer still occured and was completed in about 30 min (Fig. 4a, upper panel). In the presence of FXN, the rate of the reaction was markedly increased, with a transfer nearly complete in 5 min (Fig. 4a, lower panel). The same effect was observed with Zn-ISCU, consistent with our previous observation with as-purified ISCU (Fig. 4b)[14]. Since the rates of persulfidation of Fe-ISCU and Zn-ISCU are both much lower than the previously reported rate of persulfide formation on NFS1, persulfide transfer appears rate-limiting here too. Therefore, the effects of FXN on the rates of persulfidation of Fe-ISCU and Zn-ISCU indicate that FXN stimu-lates persulfide transfer in both cases. In contrast, FXN did not affect the rate of persulfide reduction (Fig. 4c).

Overall, these data suggest that FXN modulates the rate of Fe-S cluster assembly by acting solely on the persulfide transfer step, which is mechanistically feasible only if persulfide transfer is the rate-limiting step of the whole process. We thus compared the rate constants of persulfide transfer determined by the alkylation assays (Fig. 4a, d) with those of (i) persulfide reduction by FDX2 that is combined with formation of the [2Fe2S] cluster and determined by monitoring Fe-S cluster formation upon addition of FDX2 to a persulfidated form of the NIAU complex (Fig. 4c), and (ii) the global reaction of Fe-S cluster assembly monitored by UV-visible spectroscopy upon addition of L-cysteine (Fig. 4e), using stoichio-metric amounts of the NFS1-ISD11-ACP complex and Fe-ISCU. The values of the rate constants of persulfide transfer and Fe-S cluster assembly were comparable, both in the presence and absence of FXN, and much lower than the rate constant of persulfide reduction by FDX2, thus indicating that persulfide transfer is the rate-limiting step of the whole process (Fig. 4d–f). This provides evidence that FXN stimulates Fe-S cluster biosynthesis by accelerating persulfide transfer to Fe-ISCU.

**Zn-ISCU promotes Fe-S cluster synthesis from free sulfide.** We showed above that Zn-ISCU cannot assemble a Fe-S cluster when incubated with one equivalent of iron and L-cysteine. However, Fe-S cluster reconstitutions with as-purified ISCU which pre-sumably contained zinc, were previously reported[14–16,18,20,21]. We indeed observed reconstitution of a Fe-S cluster by Zn-ISCU in the presence of iron, but only when the concentration of L-cysteine was raised above stoichiometry (Fig. 5a). The spectro-scopic features of the Fe-S cluster generated by Zn-ISCU were identical to those of the [2Fe2S] cluster reconstituted by Fe-ISCU (Fig. 5b). Moreover, the rate of the reaction was slowed down in the absence of FXN, as reported for the reconstitutions performed with as-purified ISCU (Fig. 5c)[14,18,20]. This suggests that this reaction reproduces the phenotype of cells lacking FXN. How-ever, this reaction was about 50 times slower relative to the reaction with Fe-ISCU and required about 30 times more L-cysteine to reach the same yield in Fe-S clusters, which accounted for only 5% of L-cysteine incorporated (Fig. 5a). Furthermore, in the absence of FDX2, the reaction was only slowed down, not abolished, which suggests that this reaction does not reproduce the physiology since FDX2 is essential for Fe-S cluster biogenesis in vivo (Fig. 5c)[53,54].

We thus examined the mechanism of the reaction performed by Zn-ISCU. Since the persulfide of ISCU is not reducible in the presence of zinc (Fig. 3a), the origin of sulfide was likely NFS1. In the absence of FDX2, Fe-S clusters were still formed, which indicated that FDX2 is not the main reductant in this reaction (Fig. 5c). Instead, the dependency of the rate of the reaction on the concentration of L-cysteine suggests that L-cysteine is the

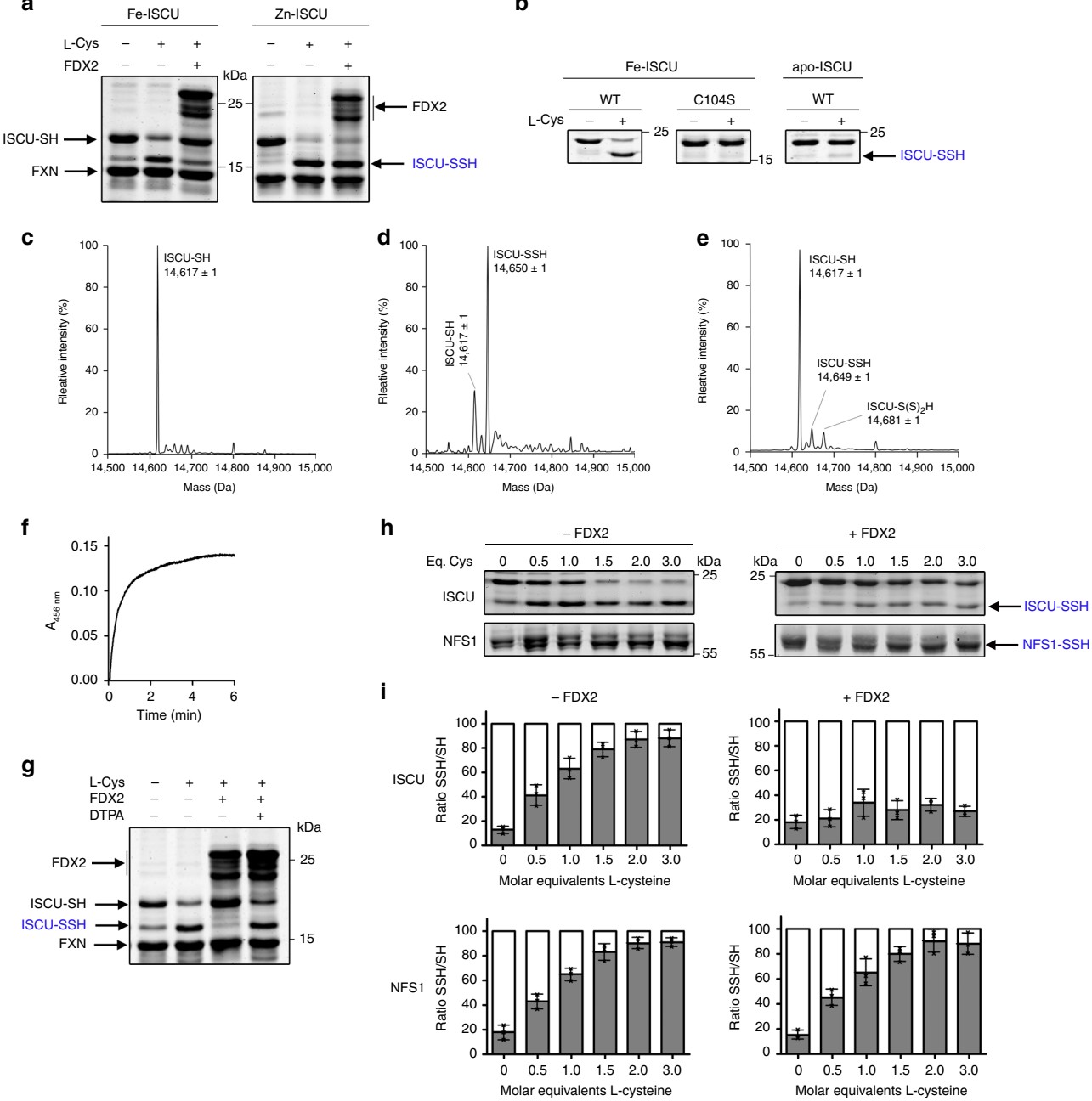

**Fig. 3** Sequence of Fe-S cluster assembly and functional role of FDX2. **a** Persulfidation of ISCU monitored by the alkylation assay in the NIAUF complex containing either Fe-ISCU or Zn-ISCU without L-cysteine, with L-cysteine and after addition of FDX2. **b** Persulfidation of the C104S mutant of Fe-ISCU and of apo-ISCU in the NIAUF complex upon addition of L-cysteine. **c–e** ESI-Q/TOF deconvoluted mass spectra of Fe-ISCU upon incubation of the NIAUF complex without (**c**) or with (**d**) L-cysteine and after Fe-S cluster reconstitution (**e**), analyzed under denaturing conditions (see Supplementary Fig. 4 for raw spectra and Supplementary Table 3 for expected masses). **f** Fe-S cluster formation monitored at 456 nm upon addition of FDX2 to the NIAUF complex containing persulfidated Fe-ISCU. **g** Effect of a metal chelator (DTPA) on persulfide reduction by FDX2. **h** Effect of FDX2 on the persulfide of NFS1 and Fe-ISCU for various amounts of L-cysteine indicated as molar ratios. **i** Quantifications of the gels in **h** expressed as the relative percentages of persulfidated (gray bar) and non-persulfidated (white bar) proteins. Error bars represent standard deviation from $n = 3$ independent experiments, uncropped gels underlying panels **a**, **b**, **g**, **h** and source data of panel **i** are provided as Source Data file

reductant (Fig. 5a). This assumption is corroborated, on one hand, by our previous data demonstrating that L-cysteine is able to reduce the persulfide of NFS1, a reaction which generates free sulfide through intermediate formation of persulfidated L-cysteine, and is stimulated by FXN, and, on the other hand, by the sigmoidal behavior observed in the absence of FDX2 that is a hallmark of the accumulation of an intermediate species (Fig. 5c)[14]. We thus propose that the poorly efficient Fe-S cluster assembly reaction

performed by Zn-ISCU proceeds via formation of free sulfide and is stimulated by FXN at the stage of sulfur transfer to L-cysteine, as previously reported with as-purified ISCU[14].

## Discussion

We show here that upon removal of its zinc ion, ISCU can bind a $Fe^{2+}$ ion in the assembly site and carry out Fe-S cluster assembly

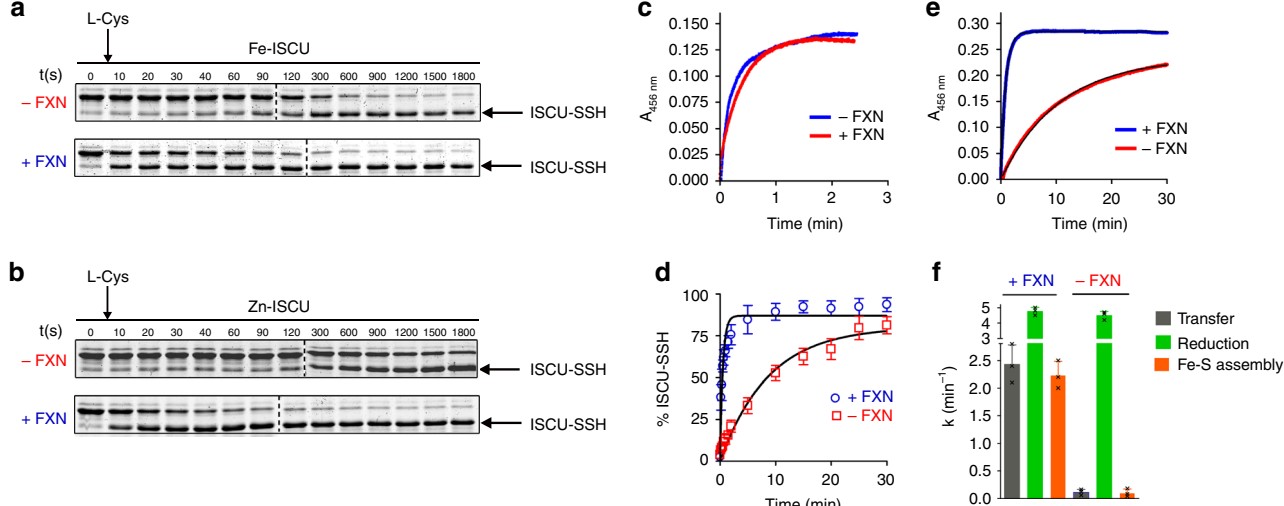

**Fig. 4** Functional role of FXN. **a**, **b** Kinetics of persulfide formation on Fe-ISCU (**a**) and Zn-ISCU (**b**) in the NIAU complex upon addition of L-cysteine, in the absence (red) and presence (blue) of FXN. **c** Kinetics of Fe-S cluster formation upon addition of FDX2 in the absence (red) and presence (blue) of FXN to the NIAU complex pre-incubated with L-cysteine. **d** Plot of percentage of persulfidated Fe-ISCU determined by quantification of the gels in **a**. The black lines represent the fits of data using a first order equation. **e** Kinetics of Fe-S cluster reconstitution under stoichiometric conditions with and without FXN. The black lines represent the fits of data using a first order equation. **f** Rate constants of persulfide transfer (gray bars), persulfide reduction combined with formation of the [2Fe2S] cluster (green bars) and global Fe-S cluster assembly upon addition of L-cysteine (orange bars), in the absence and presence of FXN, determined from data in **d**, **c**, **e** using stoichiometric amounts of the NFS1-ISD11-ACP complex and Fe-ISCU (see Methods for rate constant determination). Error bars represent standard deviation from $n = 3$ independent experiments, uncropped gels underlying panels **a**, **b** and source data of panel **d**, **f** are provided as Source Data file

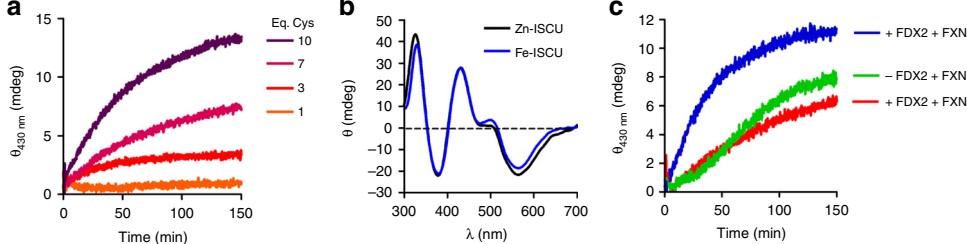

**Fig. 5** Fe-S cluster reconstitution with Zn-ISCU with excess L-cysteine. **a** Kinetics of Fe-S cluster assembly by Zn-ISCU (100 μM) monitored by CD at 430 nm, performed under catalytic conditions with the ISC components (NFS1-ISD11-ACP, FXN, FDX2, and FDXR), iron (200 μM) and various amounts of L-cysteine as indicated. **b** CD spectra of Fe-S cluster reconstitution assays by Zn-ISCU as performed in **a** with 1 mM of L-cysteine. The spectra were normalized to the spectrum obtained with Fe-ISCU. **c** Kinetics of Fe-S cluster assembly by Zn-ISCU with 10 equivalents of L-cysteine in standard conditions (blue) and in reactions missing FXN (red) or FDX2 (green);

with high efficiency, as virtually all iron and L-cysteine are incorporated in the form of [2Fe2S] clusters. Our step-by-step analysis indicates that the mechanism of assembly relies on the transfer of the persulfide of NFS1 to ISCU that is subsequently reduced into sulfide by FDX2, leading to formation of a [2Fe2S] cluster in a monomer of ISCU (Fig. 6). These data thus provide evidence that FDX2 operates in persulfide reduction and is essential since Fe-S cluster assembly was compromised in its absence. Our data also provide information on the nucleation process leading to a dinuclear [2Fe2S] cluster. As only one iron is initially present in ISCU and only one sulfide ion is generated per ISCU by FDX2, we postulate that formation of the Fe-S cluster requires dimerization of ISCU to generate a bridged [2Fe2S] cluster at the interface of the two subunits, which then segregates on one of the two monomers. Two structures of the NFS1-ISD11-ACP complex were reported which both displayed a dimeric complex but with distinct topologies[43,44]. In the structure reported by Boniecki et al., the two ISCU proteins are bound at the opposite ends of the complex thus precluding dimerization[43]. Whereas in the structure reported by Cory et al., although it does

not contain ISCU, modelling predicts that the two ISCU proteins are very close to each other, thus consistent with formation of a dimer within the complex[44].

Our data also indicate that a key feature of the Fe-ISCU based reaction is the iron-dependency of both persulfide transfer and reduction, which most likely ensure that sulfur transfer and sulfide production are coordinated with iron availability in ISCU, thereby preventing futile persulfide transfer cycles and allowing instantaneous binding of the nascent sulfide to the nearby iron. Strikingly, zinc and iron appear interchangeable for persulfide transfer. The metal ion might function as a Lewis acid creating an electrophilic character on the sulfane sulfur of the persulfide of NFS1 to facilitate the nucleophilic attack by the receptor cysteine of ISCU. In contrast, persulfide reduction by FDX2 is operative with iron, but precluded with zinc. During assembly, the iron ion switches from the +2 state in Fe-ISCU to the +3 state in the [2Fe2S]. This suggests that the iron ion is the source of one of the two electrons needed for persulfide reduction, with the other one being donated by the [2Fe2S] cluster of FDX2. This could explain why reduction does not occur on Zn-ISCU.

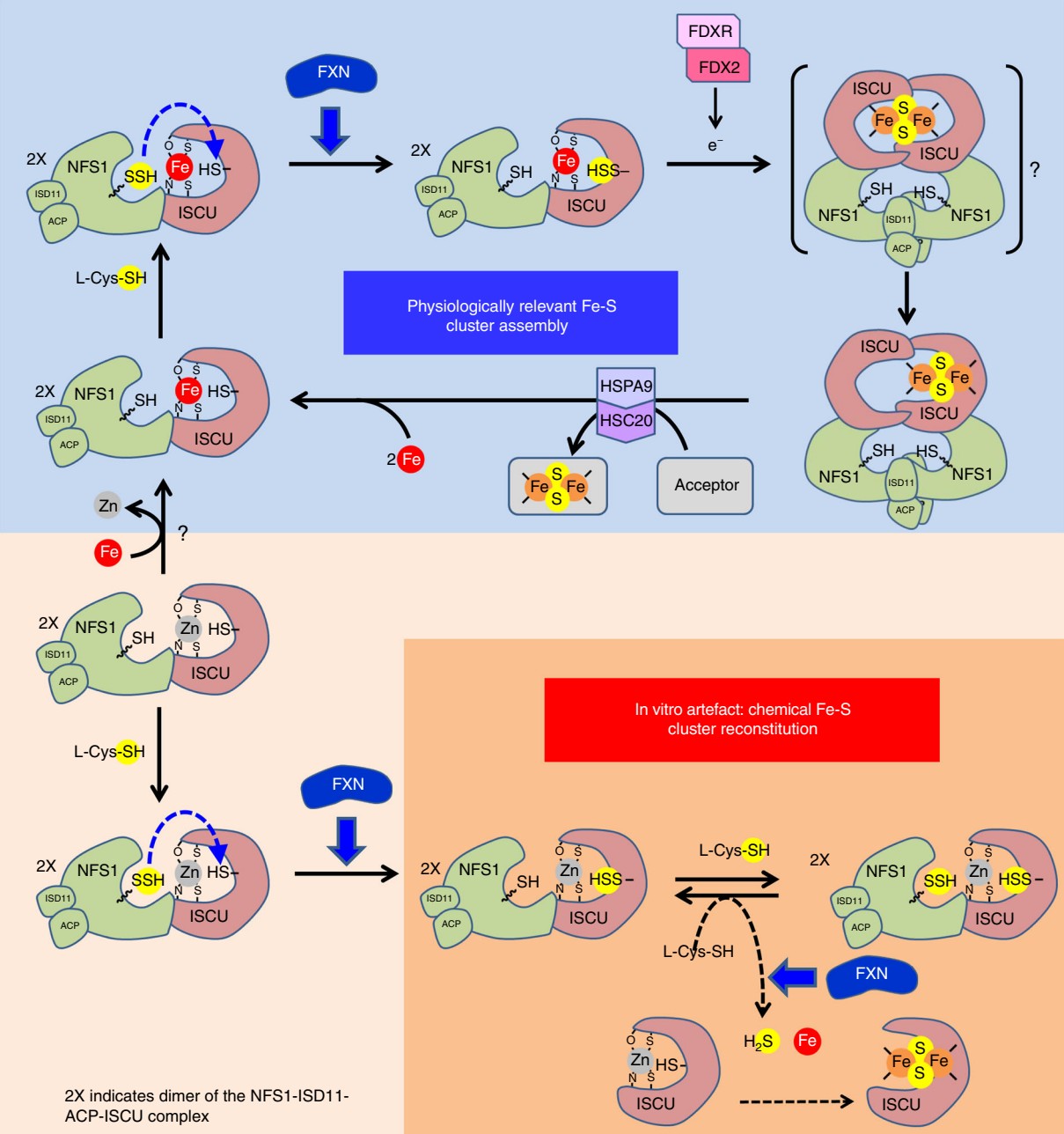

**Fig. 6** Proposed model of Fe-S cluster biosynthesis. The upper part describes physiologically relevant Fe-S cluster assembly by Fe-ISCU. Upon reaction with L-cysteine, a persulfide is generated on the catalytic cysteine of NFS1 that is transferred to the cysteine C104 of Fe-ISCU and FXN accelerates this reaction. The persulfide of Fe-ISCU is then reduced into sulfide by FDX2, which leads to formation of a [2Fe2S] cluster. The [2Fe2S] cluster is putatively formed by dimerization of ISCU that is assisted by the dimeric structure of the NIA complex. The bridged [2Fe2S] cluster subsequently segregates on one subunit and is transferred to recipient apo-proteins by the HSPA9/HSC20 chaperone system. Reloading of ISCU with iron ions, by a still ill-defined chaperone, allows subsequent turnovers. The lower part describes the reaction with Zn-ISCU that is not physiologically relevant. Upon reaction with L-cysteine, a persulfide is generated on NFS1 and is transferred to the cysteine C104 of Zn-ISCU. FXN enhances the rate of this reaction, but the persulfide of Zn-ISCU is not reduced by FDX2. Upon reaction with a second molecule of L-cysteine, the persulfide of NFS1 is regenerated. This persulfide is reduced by thiols such as L-cysteine which leads to formation of free sulfide that combines with free iron to form a Fe-S cluster in Zn-ISCU in a slow and poorly efficient process akin to chemical reconstitution (orange background). FXN accelerates the reduction of the persulfide of NFS1 by thiols

This model contrasts with the one drawn from the data obtained with Zn-ISCU (Fig. 6). When zinc is in the assembly site, the impeded reduction of the persulfide of ISCU by FDX2 fosters the reduction of the persulfide of NFS1 by L-cysteine, which leads to formation of persulfidated L-cysteine (Cys-SSH) that is decomposed into sulfide by another molecule of L-cysteine, as previously described with as-purified ISCU[14]. FDX2 also

promotes this reaction most likely by reducing Cys-SSH. The sulfide ions then combine with iron to form a Fe-S cluster that is inserted in ISCU in a poorly efficient way akin to chemical reconstitution[1,14]. This reaction might only occur in vitro, as in vivo, the persulfidated cysteine and the sulfide ions will not remain in ISCU but will diffuse throughout the cell. Moreover, the zinc ion was shown to decrease the rate of sulfide production

which may help mitigate this side-effect in vivo[46]. We thus propose that the Fe-ISCU based reconstitution is the one that recapitulates the physiological Fe-S cluster assembly process.

Since the presence of zinc has been persistently reported in eukaryotic and prokaryotic recombinant ISCU proteins, we presume that all previous Fe-S cluster reconstitutions were describing the non-physiological Fe-S cluster assembly process. Nevertheless, although the zinc ion hinders physiologically relevant Fe-S cluster assembly on ISCU, by enabling persulfide transfer to ISCU and by preventing reduction by FDX2, it might convert ISCU into a sulfur-transferase as proposed for the *Bacillus subtilis* SufU protein, a closely related homolog of ISCU[55].

Our data indicate that the function of FXN is to accelerate persulfide transfer to Fe-ISCU that is the rate-limiting step of Fe-S cluster synthesis. Further, the effect of FXN, as it does not modify the nature of the Fe-S cluster formed on ISCU and its yield, points to a regulatory function, which fits to FXN non-essential role in vivo[31,34–36]. FXN also enhances persulfide transfer to Zn-ISCU, as previously reported with as-purified ISCU[14], which might be linked to the above-proposed sulfur-transferase function of Zn-ISCU. FXN thus appears as an enhancer of metal-dependent persulfide transfer reactions. It may operate by either promoting persulfide binding to the metal center, by modifying the coordination sphere of the metal ion or by deprotonating the receptor cysteine of ISCU to increase its nucleophilic ability. Addressing these questions will be critical to understand the biochemical function of FXN.

A reconstitution of the yeast ISC machinery that included FDX2, reported that FXN is critical for Fe-S cluster synthesis, a conclusion not supported by our data[21]. In these experiments, a significant amount of Fe-S cluster (30%) was still formed in the absence of FDX2, and also in reactions lacking FXN, which are in fact typical features of the Zn-ISCU-based reaction (Fig. 5c). Therefore these data might not be used to draw any conclusions on the physiological role of FXN in Fe-S cluster biosynthesis. FXN was also proposed to deliver iron to the ISC machinery based on its ability to bind iron. However, this hypothesis has been questioned by several studies[9,29,36]. Anyhow, if indeed FXN carries this additional function, then how would zinc be exchanged? This would require another enzyme since FXN is not able to exchange zinc for iron. Our data instead indicate that FXN does not bind iron and is not required for iron insertion. Therefore, its biochemical role seems to be restricted to the stimulation of persulfide transfer.

Altogether, our data afford evidence that FXN operates in Fe-S cluster biogenesis as a kinetic activator of persulfide transfer, a function that probably is conserved in prokaryotes given the homologies of prokaryotic and eukaryotic frataxin proteins. Such a regulatory function further suggests that the expression of FXN is tightly regulated in order to adapt Fe-S cluster biogenesis to the cellular demand. Our results thus open perspectives to unravel the pivotal role of FXN in Fe-S cluster biogenesis that will help better understand the pathophysiology of FRDA and research in the development of therapeutic treatments[9].

## Methods

**Chemicals and materials**. Luria Bertani medium, protease inhibitor cocktail (sigmafast), sodium phosphate dibasic (Na$_2$HPO$_4$), urea, tris base, tris HCl, sodium dodecyl sulfate (SDS), dithiothreitol (DTT), L-cysteine, nicotinamide adenine dinucleotide phosphate (NADPH), human His$_6$-thrombin, Ethylenediamine tetraacetic acid (EDTA), diethylene triamine pentaacetic acid (DTPA), isopropyl β-D-1-thiogalactopyranoside (IPTG), ferrous ammonium sulfate (Fe(NH$_4$)$_2$(SO$_4$)$_2$), imidazole, zinc sulfate (ZnSO$_4$), 4-(2-pyridylazo)resorcinol (PAR), 3-(2-pyridyl)-5,6-diphenyl-1,2,4-triazine-p,p′-disulfonic acid monosodium salt hydrate (Ferrozine), ascorbic acid and acrylamide:bis-acrylamide (29:1) solutions were from Sigma–Aldrich. HiTrap chelating HP column (5 ml), PD-10 desalting, HiPrep

Desalting 26/10, HiLoad 16/60 prep grade Superdex 200 and Superdex 75 columns were from GE Healthcare; Amicon Ultra centrifugal filter ultracel 10 and 30 kDa from MILLIPORE, MicroBio-Spin columns from Bio-Rad; Rosetta2-(DE3) and pET28b from Novagen, cuvette from Hellma

**Protein purification**. The vectors expressing NFS1-(amino acid (aa) 59–459), ISD11, ISCU-(aa 36–168) and FXN-(aa 79–207) were kindly provided by Dr. Hélène Puccio (IGBMC, Strasbourg). His-tags followed by thrombin (Thr) cleavage sites were introduced by mutagenesis leading to the expression plasmids pCDFDuet-site1: 6xHis-Thr-NFS1-site2:ISD11, pETDuet-site1: 6xHis-Thr-ISCU and pET16-6xHis-Thr-FXN. The C35S, D37A, C61A, C96S, H103A, and C104S mutations were introduced in ISCU by site-directed mutagenesis (IMAGIF platform, Gif Sur Yvette, France). The vectors expressing human 6xHis-Thr-FDX2 (aa53-183) from pET28b was generated by GenScript (Piscataway, USA). The Rosetta2-(DE3) cells were transformed with the plasmids and the cells were grown from a single colony at 180 rpm at 37 °C in LB with appropriate antibiotics. Protein expression was induced when the cells reached OD600 = 0.6 using 500 µM IPTG and the cells incubated for an additional 16 h at 18 °C at 180 rpm for the expression of NFS1-ISD11 and 3 h at 30 °C for the expression of ISCU, FXN, and FDX2. The cell cultures were harvested by centrifugation at 5500 rpm for 10 min at 4 °C. Cell pellets were resuspended in buffer A (50 mM Na$_2$HPO$_4$, 150 mM NaCl, 5 mM imidazole, pH 8) containing a protease inhibitor cocktail. The cell suspensions were lysed by 3 cycles of French Press. Cell debris were removed by centrifugation at 45,000 rpm for 45 min at 4 °C. The His-tagged proteins NFS1-ISD11-ACP, ISCU (WT and mutants), FXN, and FDX2 were purified on a HiTrap chelating HP loaded with nickel ions. The resin was washed with buffer A and bound proteins were eluted by a linear gradient of buffer B (50 mM Na$_2$HPO$_4$, 150 mM NaCl, 500 mM imidazole, pH 8). Protein fractions were mixed and concentrated on Amicon to 4 mL. The His$_6$-NFS1-ISD11-ACP complex was incubated for 10 min with 10 mM DTT at room temperature to remove persulfide and then loaded on a HiLoad 26/600 Superdex 200 prep grade column pre-equilibrated with buffer P (50 mM Na$_2$HPO$_4$, 150 mM NaCl, pH 8). The ISCU (WT and mutants), FXN, and FDX2 proteins were desalted in buffer P on HiPrep Desalting 26/10 and the His-tags were removed by treatment with thrombin (3.5 NIH unit of His6-thrombin per mg of proteins) for 16 h at 22 °C. The ISCU proteins (WT and mutants) were incubated with 10 mM DTT for 10 min to remove persulfide. The cleaved ISCU (WT and mutants), FXN and FDX2 proteins were loaded on a HiLoad 26/600 Superdex 75 prep grade column pre-equilibrated with buffer P. All the proteins were aliquoted and stored in glycerol (10% final) at −80 °C. All these procedures were also previously described[14]. For [15]N- labeled and [13]C,[15]N-labeled ISCU, ISCU overexpressing cells were grown in M9 minimal media supplemented with [15]NH$_4$Cl (1 g.L$^{-1}$), glucose (4.g L$^{-1}$) or [13]C-glucose (2.7 g.L$^{-1}$), ampicillin (100 µg.mL$^{-1}$), chloramphenicol (34 µg.mL$^{-1}$), MgSO$_4$ (2 mM), CaCl$_2$ (50 µM), and ZnCl$_2$ (50 µM). Purified human FDXR was provided by R. Lill (Marburg). Protein concentrations were determined by UV-visible at 278 nm in urea 8 M, Tris 0.1 M, pH = 7.5 using absorption coefficients of 49965, 9700, 26335, 13850, and 44330 M$^{-1}$. cm$^{-1}$ for the NFS1-ISD11-ACP complex, ISCU, FXN, FDX2, and FDXR. The absorption of the PLP (ε = 5000 M$^{-1}$.cm$^{-1}$) was also used to double-check the concentration of the NFS1-ISD11-ACP complex.

**Metal insertion and exchange**. Zn-ISCU was prepared by incubating as-purified ISCU with 2 molar equivalents of ZnSO$_4$ followed by desalting to remove excess zinc. All iron binding properties on Zn-ISCU and preparation of Fe-ISCU were performed under anaerobic conditions using a Jacomex glove box (O$_2$ < 2 ppm). Samples were prepared by mixing 2 µL of ferrous ammonium sulfate (5 mM) prepared in H$_2$O with Zn-ISCU (100 µM), with or without FXN (100 µM) in buffer P. To prepare Fe-ISCU, the zinc ion was first removed by incubating Zn-ISCU with DTPA, followed by desalting on Superdex to remove excess DTPA. Then apo-ISCU was incubated with ferrous ammonium sulfate in various conditions as described in each section. Apo-ISCU incubated with one equivalent of ferrous ammonium sulfate is denoted Fe-ISCU thereafter.

**Fe-S cluster assembly assays**. Fe-S cluster assembly assays were performed under anaerobiosis in two different conditions: catalytic and stoichiometric. Standard catalytic conditions included Zn-ISCU (100 µM) incubated with one molar equivalent of ferrous ammonium sulfate or Fe-ISCU (100 µM), mixed with the NFS1-ISD11-ACP complex (10 µM), FXN (10 µM), FDX2 (10 µM), FDXR (1 µM), and NADPH (200 µM) in buffer P. The mix was transferred into a septum sealed cuvette and the reaction was initiated with L-cysteine (100 µM) introduced with a gastight Hamilton syringe. The kinetics of assembly were monitored by UV-visible spectroscopy at 456 nm. The kinetics of Fe-S cluster assembly with Zn-ISCU at high L-cysteine concentrations were monitored by CD at 430 nm as promiscuous Fe-S clusters were reported to be formed that could contribute to the absorption by UV-visible but not to CD[26]. Standard stoichiometric conditions included Fe-ISCU (100 µM), the NFS1-ISD11-ACP complex (100 µM), FXN (100 µM), FDX2 (10 µM), FDXR (1 µM), and NADPH (200 µM) in buffer P. The reactions were initiated by L-cysteine (100 µM) as described for catalytic conditions.

To assess the effect of FXN on the reduction of persulfide by FDX2, the reaction of persulfidation of ISCU was performed using stoichiometric amounts of Fe-ISCU

(50 μM) and the NFS1-ISD11-ACP complex (50 μM) with 1 equivalent of L-cysteine for 30 min in buffer P, then FDX2 (5 μM), FDXR (1 μM) and NADPH (100 μM) with or without FXN (50 μM) were added and the reaction of Fe-S cluster assembly was monitored at 456 nm by UV-visible spectroscopy.

**UV-visible and CD spectroscopies.** Electronic absorption spectra in the UV-visible domain were acquired using a Secomam UVIKON XL spectrometer and CD spectra using a Jasco J-815 CD spectrometer. For iron binding properties of Zn-ISCU, the samples were prepared by mixing 2 μL of ferrous ammonium sulfate (5 mM) prepared in $H_2O$ with Zn-ISCU (100 μM), with or without FXN (100 μM) in buffer P and under anaerobic conditions. For iron binding properties of apo-ISCU, the samples were prepared by mixing apo-ISCU (100 μM), WT and mutants (C35S, D37A, C61S, H103A that are devoid of zinc, and apo-C104S) prepared in buffer P with 2 μL of ferrous ammonium sulfate (5 mM) prepared in water. The mixtures were then introduced in a septum sealed cuvette.

To generate UV-visible and CD spectra of reconstituted ISCU alone and in the NIAUF complex the contribution of the proteins and non-incorporated component was subtracted. To this end, the mixture at the end of the reaction was desalted on a Biospin P6 column to remove the contribution of non-incorporated components, essentially NADPH, and the UV-visible and CD spectra were collected. Then the contribution of the proteins was removed by subtraction of the spectra of the initial mixture (minus L-cysteine and NADPH). To account for protein loss during the desalting procedure, the amount of the initial spectra to be subtracted was calculated based on total protein concentration measured by the Bradford method before and after desalting. Scales in molar absorption coefficients referred to the absorption coefficients of the [2Fe2S] clusters of free ISCU and of ISCU in the NIAUF complex (see quantifications section).

**Quantifications of zinc, iron, and Fe-S clusters.** The zinc content of ISCU was measured by inductively coupled plasma mass spectrometry (ICP-MS) at IRAMIS, CEA Saclay, France, and routinely by a colorimetric method using 4-(2-pyridylazo) resorcinol (PAR), a metallochromic indicator. In brief, a standard curve at 500 nm was prepared using zinc sulfate in the 0–10 μM range in 8 M urea, 150 mM Tris, pH 7.5 containing 100 μM of PAR. The amount of zinc in the proteins was measured in the same buffer for five different concentrations. Iron in proteins was quantified by ICP-MS. Routinely, the concentration of iron in stock solution was determined using ferrozine[56].

Fe-S clusters were quantified routinely by UV-visible spectroscopy based on absorption coefficients that were determined by quantification of iron by ICP-MS. To determine the absorption coefficient of the [2Fe2S] cluster of free ISCU and ISCU within the NIAUF complex, ISCU was reconstituted under catalytic and stoichiometric conditions, respectively, with 2 eq. of iron and L-cysteine to reach the maximum yield. For free ISCU, the proteins were separated on a Superdex 75 column under anaerobic conditions. The fractions corresponding to monomeric free ISCU holding a Fe-S cluster were pooled and a UV-visible spectrum was collected. Total iron in this sample was measured by ICP-MS. Based on iron titrations, an absorption coefficient of 7200 $M^{-1}.cm^{-1}$ was determined at 456 nm for the [2Fe2S] of free ISCU. For ISCU in the NIAUF complex, the reaction mixture was desalted on a NAP-5 column, a UV-visible spectrum was collected and total iron was measured by ICP-MS. Since FDX2 was present in this sample which contains a [2Fe2S] cluster, its contribution to total iron was subtracted based on its initial concentration. But to account for FDX2 loss during the desalting procedure, the initial amount of FDX2 was corrected by measuring total protein concentration by the Bradford method, before and after desalting. The titration of iron, corrected from the contribution from FDX2, yielded an absorption coefficient of 6000 $M^{-1}.cm^{-1}$ at 456 nm for the [2Fe2S] of ISCU in the NIAUF complex.

For titration of Fe-S clusters by iron and L-cysteine, the reconstitutions were performed under catalytic conditions with apo-ISCU (50 μM) mixed with the NFS1-ISD11-ACP complex (5 μM), FXN (5 μM), FDX2 (5 μM), FDXR (1 μM) and NADPH (100 μM) in buffer P and various amounts of iron or L-cysteine as indicated in the text, keeping the concentration of L-cysteine or iron, respectively, constant at 2 molar equivalents.

**Mössbauer spectroscopy.** Samples for Mössbauer spectroscopy were prepared by mixing 300 μL of apo-ISCU (2 mM) WT and mutants, prepared in 50 mM Tris, 150 mM NaCl, pH = 8 (buffer T), with one equivalent of $^{57}Fe$ enriched solution of ferrous ammonium sulfate (81.5 mM) prepared in $H_2O$. Samples of ISCU reconstituted with a Fe-S cluster were prepared under catalytic conditions with a mixture containing 350 μL of apo-ISCU (1 mM), $^{57}Fe$ enriched ferrous ammonium sulfate (2 mM), the NFS1-ISD11-ACP complex (5 μM), FXN (5 μM), FDX2 (5 μM), FDXR (1 μM), NADPH (2 mM), and L-cysteine (2 mM) in buffer P and after 60 min of reaction the mixture was desalted. The samples were introduced in sample holders and frozen in the glove box.

Transmission Mössbauer spectra were recorded with a conventional Mössbauer spectrometer operated in the constant acceleration mode in conjunction with a 512-channel analyzer in the time-scale mode (WissEl GmbH). The Mössbauer spectra were calibrated using α-iron foil at room temperature. A continuous flow cryostat (Optistat[DN], Oxford Instruments) was used to cool the samples to 77 K. Field-dependent conventional Mössbauer spectra were obtained with a helium

closed-cycle cryostat (CRYO Industries of America, Inc.) equipped with a superconducting magnet. The magnetic field was aligned parallel to the γ-ray beam. The spectral data were transferred from the multi-channel analyzer to a PC for further analysis employing the public domain program Vinda running on an Excel 2003® platform[57]. The spectra were analyzed by least-squares fits using Lorentzian line shapes with the linewidth Γ. Field-dependent spectra were simulated by means of the spin Hamilton formalism[58].

**Denaturing and native mass spectrometry analysis.** Nanoelectrospray ionization mass spectrometry (nanoESI-MS) analyses were carried out on a quadrupole time of flight mass spectrometer (Synapt G2 HDMS, Waters) equipped with an automated chip-based electrospray ionization technology (Triversa Nanomate, Advion) using the positive ion mode.

Prior to mass spectrometry analysis, the samples were buffer exchanged with 10 mM ammonium acetate ($AcONH_4$) at pH 7.0 using microcentrifuge gel-filtration columns (Zeba 0.5 mL, Thermo Scientific).

For denaturing MS analysis, the external calibration was performed using a 2 μM denatured horse heart myoglobin solution as a standard for denaturing analysis in the 500–5000 $m/z$ range mass spectra. Data acquisition time and scan time were set at 2-min and 4-s, respectively. Buffer exchanged samples were diluted to 10 μM in a mixture of water:acetonitrile (1:1) acidified with 1% of formic acid. Denaturing MS analyses were performed with an accelerating voltage of 40 V and the backing pressure of the instrument was fixed to 1.7 mbar. Mass spectra were deconvoluted with MassLynx 4.1 (Waters, Manchester, UK) using the MaxEnt module 1 with the following parameters: $m/z$ range: 600–3000; Gaussian smoothing: 2; mass range: 14,500–15,000 Da, number of iterations: 30; substract: 20–40%.

For native mass spectrometry analysis, cesium iodide cluster ions from a 2 g/L isopropanol:water (1:1) solution were used to calibrate the mass spectrometer up to 6000 $m/z$. Samples were diluted to 10 μM in 10 mM ammonium acetate ($AcONH_4$) at pH 7.0. Data acquisition time and scan time were set at 2-min and 4-s, respectively. To avoid the dissociation of the iron-sulfur cluster protein interaction, the analysis under native conditions were performed with an accelerating voltage of 40 V and the backing pressure of the instrument was fixed to 6 mbar. Raw data mass spectra were acquired with MassLynx 4.1 (Waters, Manchester, UK) and treated with the following parameters: $m/z$ range: 1500-4000; Gaussian smoothing: 10.

**NMR spectroscopy.** 1D $^1H$ and 2D $^1H$-$^{15}N$ SOFAST-HMQC NMR spectra of unlabeled ISCU and $^{15}N$-ISCU were recorded on a Bruker Avance III 800 MHz NMR spectrometer equipped with a TCI cryoprobe at a temperature of 293 K. Samples were in buffer P containing 7% $^2H_2O$ to lock the magnetic field. ISCU concentration was 50–100 μM. Mixed samples with NFS1-ISD11-ACP and FXN were obtained by adding 1 equivalent of each protein and concentrating the mixtures to 50 μM final concentration. Apo-ISCU and Fe-ISCU samples were in buffer T and measured in a valved tube filled in a glove box. Fe-ISCU was prepared by incubating apo-ISCU with one molar equivalent of $Fe^{2+}$. The ratio of the S to D form was estimated based on the intensities of the two Hε-Nε correlation signals of the tryptophan side chain of ISCU (denoted 74 Wε)[49].

**Maleimide-peptide alkylation assay.** Proteins were assayed for persulfide by alkylation with $MalP_{16}$[14]. The assays were routinely conducted with stoichiometric amounts of NFS1-ISD11-ACP, ISCU (WT and mutants, with and without 1 equivalent of metal) and FXN at a final concentration of 20 μM and 5 equivalents of L-cysteine. The reactions were stopped by incubation with 2 molar equivalents of $MalP_{16}$ with regard to the total concentration of thiols, under denaturing conditions (2% SDS) for 30 min; then DTT (1 mM) was added and the reactions were incubated for another 30 min before analysis by SDS-PAGE under reducing conditions. Persulfidation of ISCU and NFS1 was monitored on 14% and 10% SDS-PAGE gels, respectively.

Persulfide transfer to Fe-ISCU, Zn-ISCU, apo-ISCU, and the C104S mutant reconstituted with iron (Fe-ISCU$^{C104S}$) were assayed in the presence of FXN by incubating the proteins with the NIA complex and L-cysteine for 5 min. To assay reduction by FDX2, the persulfidated forms of Fe-ISCU and Zn-ISCU were incubated with FDX2 (20 μM), FDXR (5 μM), and NADPH (100 μM) for 5 min before analysis by the alkylation assay. To circumvent the co-migration of alkylated FDX2 with fully alkylated ISCU which corresponds to its non-persulfidated form, a mutant of ISCU lacking the cysteine residue C96, that is non-conserved, was used to decrease the number of alkylation sites and thereby decrease the size of alkylated ISCU. We have checked that the C96S mutation was silent by assessing the kinetics of Fe-S cluster assembly under catalytic conditions and by comparing the UV-visible spectra with WT ISCU (See Supplementary Fig. 5a, b). To assess the effect of iron on persulfide reduction, the persulfidated form of ISCU prepared as described above was incubated with DTPA (100 μM) for 10 min before adding FDX2 (20 μM), FDXR (5 μM), and NADPH (100 μM). We checked by UV-visible that at this concentration of DTPA the [2Fe2S] cluster of FDX2 was not destroyed and was still reducible by FDXR and NADPH (See Supplementary Fig. 5c). To assess the parallel effect of FDX2 on the persulfidation of both NFS1 and ISCU (i.e., second transfer or

direct reduction), WT Fe-ISCU (50 μM), the NIA complex (50 μM), and FXN (50 μM) were incubated for 10 min with various amounts of L-cysteine, ranging from sub-stoichiometric to excess, as indicated in the text. Then a mixture containing FDX2 (5 μM), FDXR (1 μM), and NADPH (50 μM) was added and the mixture was incubated for 10 min before analysis by the alkylation assay. The persulfidation of both, ISCU and NFS1, before and after adding FDX2, were analyzed in parallel.

To assess the effect of FXN on the kinetics of persulfide transfer, the reactions were initiated by incubating Zn-ISCU or Fe-ISCU (20 μM) and the NIA complex (20 μM) with 5 equivalents of L-cysteine in the absence or presence of FXN (20 μM). The reactions were analyzed at different time points by adding the mixture containing MalP$_{16}$ and SDS, as described above, to quench the reaction. Then the persulfidation of ISCU was analyzed as described above.

**Simulation of kinetics**. The kinetics were fitted using the enzyme-substrate model developed by Michaelis and Menten.

For catalytic conditions of reconstitution, the quasi-stationary state approximation as developed by Briggs and Haldane was applied to the linear portion of the kinetics corresponding to the steady state conditions. In this model the initial and maximum velocities, $V_i$ and $V_{max}$, are express as follows:

$$V_i = \frac{dP}{dt} = \frac{V_{max} \cdot [S_0]}{[S_0] + K_M} \tag{1}$$

$$V_{max} = k_2 \cdot [E_0] \tag{2}$$

Where P is the concentration of product, $[S_0]$ and $[E_0]$ are the initial concentrations of substrate and enzyme, respectively, $K_M$ is the Michaelis constant and $k_2$ the rate of product formation. As the initial velocities were constant at all concentrations of L-cysteine tested (Supplementary Fig. 2b), this indicated that the rate of the reaction was at its maximum velocity. The initial velocity could then be equal to the maximum velocity:

$$V_i = V_{max} = k_2 \cdot [E_0] \tag{3}$$

and after integration, the concentration of product is:

$$[P] = k_2 \cdot [E_0] \cdot t \tag{4}$$

The rate constant of Fe-S assembly, $k_2$, was extracted from the slope by linear regression of the linear portion of the curve. A similar model was applied to extract the rate constant of persulfide reduction by FDX2 by monitoring Fe-S cluster formation upon addition of FDX2 to the persulfidated form of the NIAU complex.

For stoichiometric conditions of reconstitution, the kinetics of persulfide transfer were modelled as a first-order reaction corresponding to the trans-persulfidation reaction within the NIAU complex. The curves were fitted using the following first-order equation:

$$P = P_f(1 - \exp(-k \cdot t)) \tag{5}$$

Where $P_f$ is the final concentration of product and $k$ the rate constant of persulfide transfer.

**Reporting summary**. Further information on research design is available in the Nature Research Reporting Summary linked to this article.

## Data availability
The source data underlying Figs. 1a, d, 2e, f, 3a, b, g–i, 4a, b, f are provided as a Source Data file. All other data supporting the findings of this study are available from the corresponding author on reasonable request.

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

## Acknowledgements

We are grateful to M. Tummeley for support during the record of the Mössbauer spectra and Batoul Srour for help with CD recording. We are grateful to Roland Lill's lab for providing purified ferredoxin reductase. This work was funded by the french National Research Agency (ANR-17-CE11-0021 Frataxur awarded to B. D'Autréaux, ANR AA17-PPPPCE11-0000034-02 PrxAge to M.B. Tolédano and ANR-11-LABX-0011-01 "LABEX DYNAMO"), Fondation pour la Recherche Medicale to M.B. Toledano, the MINECO (grants CTQ2014-52658-R and CTQ2017-84779-R) for peptide-maleimide synthesis, the Université de Strasbourg and the French Proteomic Infrastructure (ProFI; ANR-10-INBS-08–03) for mass spectrometry experiments. S.C. thanks GIS IBiSA and Région Alsace for financial support in purchasing a Synapt G2 HDMS instrument. T.B. acknowledges the Institut de Recherche Servier for funding of his PhD fellowship. C.S.M. and V.S. acknowledge the DFG for financial support of this work (SCHU 1251/17-1 within the SPP 1927 "Iron-Sulfur for Life").

## Author contributions

S.G. made characterization of iron containing ISCU by CD, Mössbauer and NMR, assessed efficiency of Fe-ISCU and Zn-ISCU in Fe-S cluster reconstitution assays and characterized Fe-S reconstituted ISCU by native mass spectrometry and Mössbauer. He also made L-cysteine titrations of persulfidation and reduction by FDX2 using alkylation assays. D.L. assessed the effects of FDX2 on the persulfide of Fe-ISCU and Zn-ISCU by alkylation assays and mass spectrometry, performed kinetics of persulfide transfer to Fe-ISCU and Zn-ISCU and assessed the effects of FXN on transfer. A.B.M. contributed first studies showing effect of zinc and iron in ISCU using kinetics and alkylation assays and first experiments indicating the effects of FXN on persulfide transfer. T.B. performed all the analysis by mass spectrometry and contributed interpretation of the data. C.S.M. performed all Mössbauer characterizations and simulations of these data and contributed interpretation of the results. L.P. contributed metal titrations and analysis of oligomeric states of Fe-S cluster containing ISCU. G.L.P. performed site-directed mutagenesis on ISCU. A.D.-M. contributed discussion of the results. O.B., J.A., and A.G. synthesized the maleimide-peptide. M.F. contributed interpretation of the results. V.S. contributed interpretation of Mössbauer analysis. S.C. contributed design and interpretation of the mass spectrometry experiments. C.S. designed and performed NMR analysis as well as interpretations. M.T. contributed discussion of the results and writing of the manuscript. B.D'A. designed and contributed most of the experiments, supervised S.G., D.L., A.B.M., and G.L.P., analyzed and interpreted the results and wrote the manuscript.

## Additional information

**Competing interests:** The authors declare no competing interests.

**Peer Review Information:** *Nature Communications* thanks Patricia Dos Santos, Tracey Rouault and the other anonymous reviewer for their contribution to the peer review of this work. Peer reviewer reports are available.

