## [Peer Review File · Nature Communications]

Reviewers' comments:

Reviewer #1 (Remarks to the Author):

The biological synthesis of Fe-S clusters in mitochondria is initiated by the Nfs1-Isd11-ACP cysteine desulfurase complex, which catalyzes the direct transfer of sulfur from the Cys substrate to the Fe-S cluster scaffold IscU. Prior to this study it has been reported that the rate of this sulfur transfer event is enhanced by frataxin (FNX), a protein medically relevant in the neurodegenerative and cardiac disease Friedreich's ataxia. This work provides additional mechanistic insight into this pathway by characterizing the involvement of frataxin and ferredoxin in steps involving sulfur transfer, persulfide reduction, and cluster assembly by the mammalian ISC system. While some of the results are viewed as incremental findings described by the authors and others, I consider some aspects of this report valuable contribution towards the mechanistic understanding of this essential biochemical pathway. I'm supportive of publication of this study upon consideration of the points raised below.

First, I think it is important to acknowledge previous studies leading to the current understanding in the field, which helps to understand relevance of the novel findings of this present study. As laid out, the introduction and discussion do not provide a clear narrative about what has been reported in the literature regarding the involvement of these proteins in Fe-S cluster biogenesis (see reports from Barondeau, Pastore, and Lill's groups). It has been previously established that FNX enhances the rate of sulfur transfer from Nfs1 persulfide intermediate to sulfur acceptors including IscU. It has also been demonstrated that ferredoxin, when used as a reductant, enhances the rate of cluster formation on the IscU scaffold in reactions containing frataxin. This work provides a nice validation that the initial steps in the mammalian ISC cluster biogenesis are equivalent to the already characterized steps in the yeast ISC cluster assembly (Werbert in Nat. Comm. 2014). The report expands on these analyses by narrowing the involvement of frataxin in the sulfur transfer step from Nfs1 to IscU and the involvement of ferredoxin as a reductant of persulfides on Fe-bound IscU. My recommendation is to frame both the introduction and discussion, so a non-expert reader understands what was known before and what is the new information that is provided by this study.

To me the most interesting finding is the comparison of reactions leading to cluster formation when starting from the distinct forms of IscU, namely Zn-bound versus Fe-bound forms of the scaffold. Like the reaction profile observed in yeast, the mammalian ISC system also shows that ferredoxin-dependent reduction of persulfides is only observed in the presence of Fe. The authors show that recombinantly-expressed, as-isolated IscU contains one equivalent of zinc, and that Fe-S cluster formation starting from Zn-bound IscU is mechanistically distinct from the one observed from Fe-bound IscU. This observation is relevant as most references about cluster assembly on IscU have utilized as-isolated IscU and have not considered the metal content of the scaffold prior reconstitution experiments. While both Fe-IscU and Zn-IscU adopt a structured state, assembly of clusters in reactions containing stoichiometric amounts of Cys shows that only Fe-IscU is competent of completing the synthesis of Fe-S clusters. From the results presented, it seems that the two-electron reduction of persulfide (So) to sulfide (S²⁻) requires two distinct one-electron donors. Ferredoxin is known to catalyze a one-electron transfer reactions from its redox reactive cluster.

Fe²⁺ present on Fe-IscU must undergo a one-electron oxidation to Fe³⁺ since the final product of the reaction as shown by the authors is a [2Fe-2S]²⁺ cluster bound to IscU. Thus, using this model, Zn is not able to serve as an equivalent one-electron donor in the reaction providing a mechanistic rationale for the distinct mechanisms. Based on this interpretation of the results, I tend to question the conclusion that FDX2 operates as a persulfide reductase, as its role as a reductant appears to be coupled to Fe-IscU and potentially oxidation of the metal.

From the results presented and the model proposed in Figure 6, it is not clear how the assembly of a [2Fe2S] cluster on a monomer of IscU occurs from Fe-IscU-SSH. In the structure of the NfsI-Isd11-ACP-IscU, each monomer of IscU is attached to one active site of the cysteine desulfurase. In the proposed cluster assembly scheme, each IscU coordinating one Fe would be capable of accepting one sulfur from NfsI in a reaction accelerated by FXN. The events involving persulfide reduction and cluster synthesis, however, remains to be defined. The authors suggested that two monomers of IscU carrying one Fe and one sulfur would then react to form a bridged [2Fe-2S] cluster, which then lead to segregation of the cluster to one monomer. At which point ferredoxin acts is not established; the results indicate that ferredoxin accelerates the rate of Fe-S formation, but it doesn't indicate how. Perhaps figure 6 could be modified to indicate this uncertainty.

Minor points

Figure 2A and B- It is not obvious the concentration of Fe used in the reconstitution experiments with Zn-IscU. One may guess that the concentration was the same as IscU, but this information is not clear from the method description. To that instance, have the authors attempted the similar experiment for Zn-IscU as the one described on panels E and F?

Figure 4 – I had a hard time understanding how the rate of persulfide reduction could be deconvoluted from the rate of cluster assembly since both reactions detected the amount of cluster accumulated on IscU. How the rate of persulfide reduction was measured (panel F)? Are the authors comparing the rates of cluster assembly under FDX limiting and saturating conditions? Perhaps additional information will be helpful to readers like me. :-)

Figure 5A – Indicate the amount of Cys equivalents considered as excess.

reviewed by Patricia Dos Santos

Reviewer #2 (Remarks to the Author):

FXN has been proposed to operate as an iron storage protein, an iron chaperone, a gatekeeper, a sulfide production stimulator and an enhancer of persulfide transfer in this process. In yeast, the homologue of FXN, Yfh1, was proposed to stimulate the formation of persulfide on Nfs1 by increasing the affinity of L-cysteine for Nfs1 (Pandey et al JBC 2013). In contrast, in a previous publication from the authors of the current manuscript, Parent et al reported that mammalian FXN is not required for persulfide formation on NFS1 but enhances the rate of persulfide transfer to ISCU (Parent et al Nat Comm. 2015). In the current manuscript, the authors provided further evidence that FXN accelerate persulfide transfer.

To give the readers a better understanding of the background and the progress made in the current study, the introduction should be clearer about the different steps being examined in this study and the previous studies (i.e. persulfide formation on NFS1, persulfide transfer from NFS1 to ISCU and persulfide reduction on ISCU).

The authors should discuss briefly Pandey et al. 2013 and Parent et al. 2015 to give readers a better understanding of the background and to help clarify the data shown in Fig 4A (as discussed below).

Overall the experiments are well-designed, and clear results and the main conclusions are based on experimental evidence. Some major points and one minor comment need to be addressed:

Major points:

1. Page.4: in describing the results on the purification of murine ISCU, which yielded a protein containing zinc, but not iron, the authors need to reference, in addition to refs. 16-18, the study from Fox and colleagues (Biochimie 152 (2018) 211-218), which also reported that recombinant human ISCU purified from *E. coli* bound Zn(II).
2. Consistent with the data presented in the present manuscript, the work from Fox and colleagues (Biochimie 152 (2018) 211-218) reported a regulatory role for Zn-bound ISCU in modulating the cysteine desulfurase activity of human NFS1, and identified Cys69 and Cys95, but not Cys138 of the human protein, corresponding to C35 and C61, and C104, respectively, of the murine Iscu (numbering of mature protein without the mitochondrial leader sequence), as the zinc ligating residues. The study from Fox and colleagues needs to be acknowledged.
3. Figure 3: using a previously validated protein persulfidation assay, the authors monitored persulfide transfer from NFS1 to ISCU, and confirmed transfer of a single sulfur on ISCU by mass spectrometry. Addition of FDX2 to the persulfidated form of Fe-ISCU led to disappearance of the persulfide and formation of an Fe-S cluster. Overall, the experiments were well designed and included all the appropriate controls. However,

the authors need to perform and present a quantification of the data in figure 3h. It is in fact necessary to show the amounts of persulfidated ISCU and NFS1 vs non-persulfidated ISCU and NFS1 at the different substoichiometric concentrations of L-cysteine employed in the assay to be able to accurately interpret the data, as the immunoblots presented aren't sufficiently clear and self-

explanatory. The authors also need to provide technical details on how the experiment of panel 3h was performed. My interpretation, after carefully reading the method section, is that they used the persulfidated form of Fe-ISCU in this assay and tested the effect of FDX2 on persulfide formation on NFS1 at substoichiometric concentrations of cysteine. This information needs to be present in the main text in order to introduce the assay and to properly interpret the results.

4. Figures 4a and 4b present groupings of lanes likely from different parts of the same gel or from different gels (the latter case would be unacceptable) which have not been made explicit by the use of dividing lines. The authors need to correct this issue which is not in adherence with the guidelines on allowed image manipulations.

5. The discussion is very poorly written and needs to be expanded in order to incorporate some essential considerations. Based on their data, the authors ruled out a potential role of FXN as an iron chaperone protein. It could well be that FXN operates as an enhancer of persulfide transfer, as proposed here. However, the authors report that “in the absence of metal (i.e. iron), persulfide transfer (from NFS1 to ISCU) was abolished, which indicates that the metal iron is required for persulfide transfer (Fig. 3b, Suppl. Figure 3 and Suppl. Text 3)”. The assays presented here were performed *in vitro* in the presence of Fe-bound ISCU. The authors need to discuss how they think ISCU gets the iron *in vivo*. It is, in fact, at least conceivable that FXN may exert a dual role by providing iron and enhancing the rate of persulfide transfer from NFS1 to ISCU. In Boniecki et al. (Nat. Comm. 2017), the crystal and SAXS data on the NFS1/ISD11/ACP/FXN complex showed that the acidic patches of FXN, which were reported to bind iron, were located at the interface of the NFS1/ISCU complex and in close proximity with the persulfide loop of NFS1. Is it then possible that, by providing iron, FXN enhances persulfide transfer from NFS1 to ISCU? Any thought on the mechanism by which FXN enhances persulfide transfer?

Does the purified frataxin used in the present manuscript bind any metal? Can the authors include ICPMS results on potential metal binding to their purified FXN?

In their model the authors show an ISCU monomer carrying a 2Fe-2S cluster, which is consistent with the reported crystal structure of a dominant negative mutant of Iscu from *Azotobacter vinelandii* in which Asp37 had been replaced by alanine, which trapped the 2Fe2S cluster, normally very labile, on the scaffold IscU. However, on page 11, they state that “formation of a [2Fe2S] cluster likely requires dimerization of ISCU”, which implies that the cluster is half-formed on two ISCU molecules of a dimeric NFS1/ACP/ISD11/ISCU complex and subsequently transferred to one of the two ISCU subunits. This conclusion would be consistent only with the work by Cory et al (reference 8), but not with reference 9, which refers to the work by Boniecki et al (Nat. Comm 2017), in which the two ISCU molecules are at the opposite ends of the NFS1/ISD11/ACP complex.

As a minor point, the authors need to make sure the nomenclature for the proteins is correct throughout the manuscript. For instance, they should use “Iscu” for the murine protein and not ISCU, which refers to the human nomenclature, as reported in the text.

Also, in the abstract, Abstract: the authors wrote “promotes Fe-S cluster synthesis from sulfide leakage”.

It is not clear what the authors mean here. The term sulfide leakage was better explained in p. 3, but the sentence was unclear in the abstract. Please clarify and rephrase.

p. 7 and Table 1: The authors wrote, “The lack of the minor species (component 2) in the samples of the H103A and C104S mutants further suggested that H103 is a ligand of this iron [component 2]”

If so, then one might expect to see only component 2 and no component 1 in C35A, D37A, or C61A mutants? Do the authors have experimental data on these mutants?

- p.7: The authors wrote, “When FXN was omitted, the assembly was slowed down, consistent with the significant decrease of Fe-S cluster biogenesis activity in cells lacking FXN (Fig. 2g).”

Fig 2g only showed the time course up to 20 min. Did you look at longer than 20 min?

In other words, in addition to slowing down, does the reaction in the absence of FXN go to completion with 90% of reconstituted ISCU?

Supplementary Fig 3 and Text 3: the authors wrote “The 1D 1H NMR spectra of apo-ISCU and Zn-ISCU show an almost complete disappearance of resonances upon mixing with one molar equivalent of the NIA complex with and without FXN (see Supplementary Fig. 3)... [which] is attributed to complete broadening of the signals and thus provides evidence for complex formation.

The reviewer agrees with the authors on this point with respect to apo-ISCU but not fully with respect to Zn-ISCU. The resonances of Fxn in the presence of apo-ISCU + NIA are almost completely disappeared, but only “half” disappeared in Fxn + Zn-ISCU + NIA. Can the authors comment on this apparent difference between apo-ISCU and Zn-ISCU? Do they have different binding affinity to NIA?

- P.9: The authors wrote: “ In the absence of metal, persulfide transfer was abolished, despite formation of complexes between apo-ISCU, FXN and the NIA complex, which indicates that the metal ion is required for persulfide transfer (Fig. 3b, Supplementary Fig. 3 and Supplementary Text 3).

It might be clearer if the sentence reads “In the absence of metal, persulfide transfer was abolished (Fig. 3b), despite formation of complexes between apo-ISCU, FXN and the NIA complex (Supplementary Fig. 3 and Supplementary Text 3)), which indicates that the metal ion is required for persulfide transfer.

- Fig 4A:

For readers who were not familiar with Parent et al 2015, the lower panel in Fig 4A might be alternatively interpreted as evidence that FXN increased persulfide formation and therefore increased formation of ISCU-SSH. Thus, it is important to provide that background information in the introduction that Parent et al. 2015 have already reported that mammalian FXN is not required for persulfide formation on NFS1.

Reviewer #3 (Remarks to the Author):

This paper addresses the mechanism of [2Fe-2S] cluster formation on the ISCU scaffold protein. While the work does advance the pool of experimental data that provides mechanistic insight on this process, several key observations are already generally known, while the principal conclusions do not move the mechanistic understanding to the extent one might expect of a Nature publication.

It is known that bacterially expressed protein is often isolated in the zinc bound form. In fact, early crystal structures were usually reported in the zinc bound state. Of course, this does not mean that under physiological concentrations and conditions, and in murine or human cells, that the protein actually exists with bound, which is relatively low in available concentration. The point that iron is also ligated to O/N ligands is also recognized through prior Mössbauer and EXAFS studies.

While the authors dismiss the role of frataxin in iron delivery, there is still a need to deliver iron, because free iron will not be found in a cell. While frataxin has been shown to serve a number of other roles (the specifics depending on the organism), nevertheless it is required to complex with ISCU and other ancillary proteins, most likely modulates the activities of other partners, but is still capable of delivering iron. If not frataxin, then how is iron delivered in the scheme presented in Fig. 6 of the manuscript? The manuscript spends quite a bit of time discussing sulfur delivery, but how we go from no sulfide to two sulfide is not clearly defined by the experimental work described in the text and in Fig. 6. The proposed ISCU dimer formation as a vehicle for bringing together two “Fe-S” pieces is rather speculative and not well supported by experimental data.

Overall, the reported work provides incremental advances, perhaps helps to clarify some prior published work, but certainly does not reveal the roles of Fd2 and frataxin in persulfide processing and does not provide a clearer definition of cluster assembly mechani

Reviewers' comments:

Reviewer #1 (Remarks to the Author):

We thank the reviewer for all the positive comments and criticisms, which helped us improve the quality of the manuscript.

Major Points:

Point 1: My recommendation is to frame both the introduction and discussion, so a non-expert reader understands what was known before and what is the new information that is provided by this study.

Action: We have now extended the introduction and the discussion to include background information of previous data. We emphasize in the discussion, as pointed by the reviewer, that all previous reconstitutions presumably used a non-physiological Fe-S cluster assembly reaction, due to the presence of zinc and the use of excess of L-cysteine, or were contaminated with this reaction (since zinc was not systematically measured, we cannot make here a definitive statement). These reconstitutions include the report by Werbert et al. in Nat. Comm. 2014, in which the non-physiological nature of the reconstitution is evidenced by the significant Fe-S background observed in the absence of FDX2 (30%, Fig.1c) that is typical of the Zn-based reaction, not the Fe-based one.

Point 2: From the results presented, it seems that the two-electron reduction of persulfide (So) to sulfide (S²⁻) requires two distinct one-electron donors. Ferredoxin is known to catalyze a one-electron transfer reactions from its redox reactive cluster. Fe²⁺ present on Fe-IscU must undergo a one-electron oxidation to Fe³⁺ since the final product of the reaction as shown by the authors is a [2Fe-2S]²⁺ cluster bound to IscU. Thus, using this model, Zn is not able to serve as an equivalent one-electron donor in the reaction providing a mechanistic rationale for the distinct mechanisms. Based on this interpretation of the results, I tend to question the conclusion that FDX2 operates as a persulfide reductase, as its role as a reductant appears to be coupled to Fe-IscU and potentially oxidation of the metal.

Answer: The reviewer is right by pointing out that FDX2 is not exactly a persulfide reductase since electrons from the Fe(II) centre are probably needed to complete the reaction.

Action: We have removed this statement throughout the manuscript. We also provide in the discussion, hypotheses on the mechanism of reduction of the persulfide.

Point 3: From the results presented and the model proposed in Figure 6, it is not clear how the assembly of a [2Fe2S] cluster on a monomer of IscU occurs from Fe-IscU-SSH. In the structure of the NfsI-Isd11-ACP-IscU, each monomer of IscU is attached to one active site of the cysteine desulfurase. In the proposed cluster assembly scheme, each IscU coordinating one Fe would be capable of accepting one sulfur from NfsI in a reaction accelerated by FXN. The events involving persulfide reduction and cluster synthesis, however, remains to be defined. The authors suggested that two monomers of IscU carrying one Fe and one sulfur would then react to form a bridged [2Fe-2S] cluster, which then lead to segregation of the cluster to one monomer. At which point ferredoxin acts is not established; the results indicate that ferredoxin accelerates the rate of Fe-S formation, but it doesn't indicate how. Perhaps figure 6 could be modified to indicate this uncertainty.

Answer: Based on the quantification of iron and sulfide provided to ISCU, we indeed postulate that the [2Fe2S] cluster is formed at the interface of an ISCU dimer, therefore that a bridged [2Fe2S] cluster should exist after persulfide reduction. However, as pointed by the reviewer, it remains to be

determined when the dimer is formed, before persulfide transfer and stimulation by FXN, before reduction by FDX2 or after?

The reviewer also says: “the results indicate that ferredoxin accelerates the rate of Fe-S formation, but it doesn’t indicate how”. Perhaps the reviewer means FXN? FDX2 does not accelerate the reaction, but is critical to it.

Action: It is difficult to indicate all these hypotheses on the same scheme of Fig. 6. For the sake of clarity, we have introduced a putative dimer of ISCU after persulfide reduction by FDX2, which is the only one for which we can reasonably postulate the existence. As also suggested by reviewer 2, we now provide a discussion on the mechanism of stimulation of persulfide transfer by FXN.

Minor points:

Minor point 1: Figure 2A and B- It is not obvious the concentration of Fe used in the reconstitution experiments with Zn-IscU. One may guess that the concentration was the same as IscU, but this information is not clear from the method description.

Answer: In both Fig. 2A and B experiments, one equivalent of iron was used with Zn-ISCU.

Action: We have clarified this point in the legend.

Minor point 2: To that instance, have the authors attempted the similar experiment for Zn-IscU as the one described on panels E and F?

Answer: Similar titrations by L-cysteine were performed with Zn-ISCU and are presented in Fig. 5b.

Minor point 3: Figure 4 – I had a hard time understanding how the rate of persulfide reduction could be deconvoluted from the rate of cluster assembly since both reactions detected the amount of cluster accumulated on IscU. How the rate of persulfide reduction was measured (panel F)? Are the authors comparing the rates of cluster assembly under FDX limiting and saturating conditions? Perhaps additional information will be helpful to readers like me. :-)

Answer: The rate constants of persulfide transfer and Fe-S cluster assembly were measured in two distinct experiments. The kinetics of persulfide transfer were assessed using the alkylation assay in reaction without FDX2, to prevent reduction of the persulfide on ISCU (Fig. 4a). The gels were quantified and the rate constant were determined by simulating these data, assuming a 1st order kinetic (Fig. 4d). The rate constants of Fe-S cluster assembly were determined by fitting the UV-visible trace of reactions performed under the conditions of the alkylation assays, i.e. with stoichiometric amounts of the NFS1-ISD11-ACP complex and Fe-ISCU, and in this case, in the presence of FDX2, to allow the reaction to proceed until formation of the Fe-S cluster.

Action: A detailed description of how these rate constants were determined has been added to the text.

Minor point 4: Figure 5A – Indicate the amount of Cys equivalents considered as excess.

Action: This has been indicated

reviewed by Patricia Dos Santos

Reviewer #2 (Remarks to the Author):

We thank the reviewer for all the constructive comments that have helped us to improve the manuscript.

Major points:

Point 1. To give the readers a better understanding of the background and the progress made in the current study, the introduction should be clearer about the different steps being examined in this study and the previous studies (i.e. persulfide formation on NFS1, persulfide transfer from NFS1 to ISCU and persulfide reduction on ISCU). The authors should discuss briefly Pandey et al. 2013 and Parent et al. 2015 to give readers a better understanding of the background and to help clarify the data shown in Fig 4A (as discussed below).

Answer: This point was also raised by reviewer 1

Action: We now provide in the introduction a comprehensive review of the literature. We have also mentioned the paper by Pandey et al. (see below).

Point 2. Page.4: in describing the results on the purification of murine ISCU, which yielded a protein containing zinc, but not iron, the authors need to reference, in addition to refs. 16-18, the study from Fox and colleagues (Biochimie 152 (2018) 211-218), which also reported that recombinant human ISCU purified from E. coli bound Zn(II).

Action: This reference has been added

Point 3. Consistent with the data presented in the present manuscript, the work from Fox and colleagues (Biochimie 152 (2018) 211-218) reported a regulatory role for Zn-bound ISCU in modulating the cysteine desulfurase activity of human NFS1, and identified Cys69 and Cys95, but not Cys138 of the human protein, corresponding to C35 and C61, and C104, respectively, of the murine Iscu (numbering of mature protein without the mitochondrial leader sequence), as the zinc ligating residues. The study from Fox and colleagues needs to be acknowledged.

Answer: We agree with the reviewer that the paper by Fox et al. reports an interesting effect of zinc on the production of sulfide by NSF1. This work also confirms the coordination of the zinc ion in the human complex.

Action: We have thus referenced this work in the manuscript. Since Fox et al measured production of sulfide in the presence of DTT, the reaction they reported corresponds to the non-physiological reconstitution in the presence of zinc. The zinc-based reaction is operative by virtue of the reduction of the persulfide of NFS1 by DTT, and not the persulfide of ISCU that is hardly DTT reducible. We have acknowledged this work in the discussion part dedicated to Fe-S cluster assembly with Zn-ISCU.

Point 4. Figure 3: using a previously validated protein persulfidation assay, the authors monitored persulfide transfer from NFS1 to ISCU, and confirmed transfer of a single sulfur on ISCU by mass spectrometry. Addition of FDX2 to the persulfidated form of Fe-ISCU led to disappearance of the persulfide and formation of an Fe-S cluster. Overall, the experiments were well designed and included all the appropriate controls. However, the authors need to perform and present a quantification of the data in figure 3h. It is in fact necessary to show the amounts of persulfidated ISCU and NFS1 vs non-persulfidated ISCU and NFS1 at the different substoichiometric concentrations of L-cysteine employed in the assay to be able to accurately interpret the data, as the immunoblots presented aren't sufficiently clear and self-explanatory. The authors also need to provide technical details on how the experiment of panel 3h was performed.

Action: We have added quantifications of the coomassie stained gels to provide the percentages of the persulfidated and non-persulfidated forms of NFS1 and ISCU (Fig. 3i), as well as more technical details of the assay in the text.

Point 5. My interpretation, after carefully reading the method section, is that they used the persulfidated form of Fe-ISCU in this assay and tested the effect of FDX2 on persulfide formation on NFS1 at substoichiometric concentrations of cysteine. This information needs to be present in the main text in order to introduce the assay and to properly interpret the results.

Answer: The reviewer says “tested the effect of FDX2 on persulfide formation on NFS1”: however, FDX2 is not involved in the formation of the persulfide of NFS1. Therefore, the reviewer probably meant testing the effect of FDX2 on persulfide reduction? These experiments were performed by incubating reaction mixtures containing the NFS1-Fe-ISCU complex with various amounts of L-cysteine, from substoichiometric amounts to excess, to generate the persulfide on NFS1 and ISCU, then a mixture containing FDX2, FDXR and NADPH was added. The persulfidation state of NFS1 and ISCU in these reaction mixtures was then analysed by the alkylation assay.

Action: these information now appear in the main text.

Point 6. Figures 4a and 4b present groupings of lanes likely from different parts of the same gel or from different gels (the latter case would be unacceptable) which have not been made explicit by the use of dividing lines. The authors need to correct this issue which is not in adherence with the guidelines on allowed image manipulations.

Answer: We indeed constructed these figures from different gels, but we did not seek to hide this fact as we left a small blank between the two gels to indicate that they were not from the same experiment.

Action: As requested by the reviewer, we have now added a dividing line between the different gels.

Point 7. The discussion is very poorly written and needs to be expanded in order to incorporate some essential considerations. Based on their data, the authors ruled out a potential role of FXN as an iron chaperone protein. It could well be that FXN operates as an enhancer of persulfide transfer, as proposed here. However, the authors report that “ in the absence of metal (i.e. iron), persulfide transfer (from NFS1 to ISCU) was abolished, which indicates that the metal iron is required for persulfide transfer (Fig. 3b, Suppl. Figure 3 and Suppl. Text 3)”. The assays presented here were performed in vitro in the presence of Fe-bound ISCU. The authors need to discuss how they think ISCU gets the iron in vivo. It is, in fact, at least conceivable that FXN may exert a dual role by providing iron and enhancing the rate of persulfide transfer from NFS1 to ISCU. In Boniecki et al. (Nat. Comm. 2017), the crystal and SAXS data on the NFS1/ISD11/ACP/FXN complex showed that the acidic patches of FXN, which were reported to bind iron, were located at the interface of the NFS1/ISCU complex and in close proximity with the persulfide loop of NFS1. Is it then possible that, by providing iron, FXN enhances persulfide transfer from NFS1 to ISCU?

Answer: The reviewer suggests that FXN could enhance persulfide transfer from NFS1 to ISCU by providing iron. However, by itself, iron does not stimulate this process. Stimulation of persulfide transfer and iron insertion appear to us as two distinct events. Iron is required for persulfide transfer, and FXN for the stimulation of this iron-dependent reaction. Consequently, in the absence of FXN, we should not observe any persulfide transfer if FXN was providing iron to ISCU, which is not the case.

The reviewer also suggests that FXN could exert a dual role: stimulation of persulfide transfer and iron chaperone, which is indeed conceivable. However, such an essential function (iron delivery) is

not consistent with the dispensable role of FXN *in vivo* in Fe-S cluster biogenesis, whereas a regulatory role in persulfide transfer appears in agreement with this ancillary function. Moreover, we could not detect any iron in purified FXN (see below) and our iron binding assays showed that FXN is not required for iron insertion, which further invalidates the iron chaperone hypothesis. Finally, the stimulatory effect of FXN on persulfide transfer fully recapitulates its global impact on Fe-S cluster biogenesis. Altogether, our data are thus not in favour of the iron chaperone hypothesis.

Action: We have now expanded the discussion, especially with regards to the iron chaperone hypothesis.

Point 8. Any thought on the mechanism by which FXN enhances persulfide transfer?

Answer: This is a good but very challenging question we would like to address in the future. For now, we have provided a few hypotheses in the conclusion.

Point 9. Does the purified frataxin used in the present manuscript bind any metal? Can the authors include ICPMS results on potential metal binding to their purified FXN?

Action: We now include metal titrations experiments which show that FXN does not contain detectable amounts of iron or zinc, which further invalidates the iron chaperone hypothesis.

Point 10. In their model the authors show an ISCU monomer carrying a 2Fe-2S cluster, which is consistent with the reported crystal structure of a dominant negative mutant of Iscu from *Azotobacter vinelandii* in which Asp37 had been replaced by alanine, which trapped the 2Fe2S cluster, normally very labile, on the scaffold IscU. However, on page 11, they state that “formation of a [2Fe2S] cluster likely requires dimerization of ISCU”, which implies that the cluster is half-formed on two ISCU molecules of a dimeric NFS1/ACP/ISD11/ISCU complex and subsequently transferred to one of the two ISCU subunits. This conclusion would be consistent only with the work by Cory et al (reference 8), but not with reference 9, which refers to the work by Boniecki et al (Nat. Comm 2017), in which the two ISCU molecules are at the opposite ends of the NFS1/ISD11/ACP complex.

Answer: The reviewer is correct: our conclusion is supported by the structure of Cory et al. but not by the structure reported by Boniecki et al. We hypothesize that these two structures may correspond to two different conformations of the same complex.

Action: We now explain this hypothesis with more details in the discussion.

Minor points:

Minor point 1. The authors need to make sure the nomenclature for the proteins is correct throughout the manuscript. For instance, they should use “Iscu” for the murine protein and not ISCU, which refers to the human nomenclature, as reported in the text.

Answer: The nomenclature with upper-case letters applies to all mammals, thus including mouse (see Colin et al. JACS 2013). The nomenclature with only the first letter upper-cased applies to yeast.

Minor point 2. Also, in the abstract, Abstract: the authors wrote “promotes Fe-S cluster synthesis from sulfide leakage”. It is not clear what the authors mean here. The term sulfide leakage was better explained in p. 3, but the sentence was unclear in the abstract. Please clarify and rephrase.

Answer: We have removed leakage from the abstract and replaced it by “free sulfide”.

Minor point 3. p. 7 and Table 1: The authors wrote, “The lack of the minor species (component 2) in the samples of the H103A and C104S mutants further suggested that H103 is a ligand of this iron [component 2]”. If so, then one might expect to see only component 2 and no component 1 in C35A, D37A, or C61A mutants? Do the authors have experimental data on these mutants?

Answer: We indeed present CD spectra of the C35A, D37A and C61A mutants showing complete lack of component 1 (Fig. 1f). The Mössbauer parameters of component 2 indicate that this species is bound by N/O ligands but no cysteine. We thus expect to see exclusively component 2 in the mutants of the cysteines: C35A and C61A, and possibly none of these components (1 or 2) in the D37A mutant as D37 might be a ligand in component 2. We hope to address this interesting question in a future study.

Minor point 4. p.7: The authors wrote, “When FXN was omitted, the assembly was slowed down, consistent with the significant decrease of Fe-S cluster biogenesis activity in cells lacking FXN (Fig. 2g).” Fig 2g only showed the time course up to 20 min. Did you look at longer than 20 min? In other words, in addition to slowing down, does the reaction in the absence of FXN goes to completion with 90% of reconstituted ISCU?

Action: We have replaced Fig. 2g by a new experiment which extends the kinetics up to 50 min, together with a CD spectrum at the end of the reaction (Supplementary Fig. 2d). These data show that the yields of Fe-S clusters are similar in reactions containing or not FXN, albeit with a slightly lower amount produced in the reaction missing FXN. The CD spectra of the Fe-S cluster formed in the absence and presence of FXN are the same, which strengthen the idea that FXN only accelerates Fe-S cluster assembly, but does not modify the reaction product. We thank the reviewer for suggesting to incorporate these data.

Minor point 5. Supplementary Fig 3 and Text 3: the authors wrote “The 1D 1H NMR spectra of apo-ISCU and Zn-ISCU show an almost complete disappearance of resonances upon mixing with one molar equivalent of the NIA complex with and without FXN (see Supplementary Fig. 3)... [which] is attributed to complete broadening of the signals and thus provides evidence for complex formation. The reviewer agrees with the authors on this point with respect to apo-ISCU but not fully with respect to Zn-ISCU. The resonances of Fxn in the presence of apo-ISCU + NIA are almost completely disappeared, but only “half” disappeared in Fxn + Zn-ISCU + NIA. Can the authors comment on this apparent difference between apo-ISCU and Zn-ISCU? Do they have different binding affinity to NIA?

Answer: The reviewer is right, it appears that FXN has a stronger affinity for the NFS1-ISCU-ACP-ISCU complex in the absence of metal in ISCU, but the reason of this effect is unclear at the moment. The structure of the human NFS1-ISCU-ACP-ISCU from Boniecki et al. show subtle changes upon zinc binding, essentially the loop Ala41-Cys44 (corresponding to Ala32-Cys35 in murine numbering) in the environment of the metal ion that is moving, which could explain this effect, but we do not have more evidence.

Action: We have added a comment in the supplementary text 3 on this matter.

Minor point 6. P.9: The authors wrote: “ In the absence of metal, persulfide transfer was abolished, despite formation of complexes between apo-ISCU, FXN and the NIA complex, which indicates that the metal ion is required for persulfide transfer (Fig. 3b, Supplementary Fig. 3 and Supplementary Text 3). It might be clearer if the sentence reads “In the absence of metal, persulfide transfer was abolished (Fig. 3b), despite formation of complexes between apo-ISCU, FXN and the NIA complex (Supplementary Fig. 3 and Supplementary Text 3)), which indicates that the metal ion is required for persulfide transfer.

Action: This sentence has been modified accordingly

Minor point 7. Fig 4A: For readers who were not familiar with Parent et al 2015, the lower panel in Fig 4A might be alternatively interpreted as evidence that FXN increased persulfide formation and therefore increased formation of ISCU-SSH. Thus, it is important to provide that background information in the introduction that Parent et al. 2015 have already reported that mammalian FXN is not required for persulfide formation on NFS1.

Action: We have now added this information in the introduction and in the corresponding part of the results section.

Reviewer #3 (Remarks to the Author):

This paper addresses the mechanism of [2Fe-2S] cluster formation on the ISCU scaffold protein. While the work does advance the pool of experimental data that provides mechanistic insight on this process, several key observations are already generally known, while the principal conclusions do not move the mechanistic understanding to the extent one might expect of a Nature publication.

Answer: We thank the reviewer for her/his comments, but we have to disagree on her/his general conclusions. We managed here to isolate for the first time an iron-loaded form of ISCU with iron in the assembly site, which assembles Fe-S cluster through persulfide transfer and FDX2 dependent persulfide reduction. This result has never been obtained in previous studies of this process, due to the presence of a zinc ion that, we show here, has hindered iron binding. We further provide a complete mechanistic description of the process of sulfide production involving FDX2.

It is known that bacterially expressed protein is often isolated in the zinc bound form. In fact, early crystal structures were usually reported in the zinc bound state. Of course, this does not mean that under physiological concentrations and conditions, and in murine or human cells, that the protein actually exists with bound, which is relatively low in available concentration.

The point that iron is also ligated to O/N ligands is also recognized through prior Mössbauer and EXAFS studies.

Answer: The reviewer might have overlooked some crucial parts of our work. First, to the best of our knowledge, previous studies only reported on the characterization of the Fe-S cluster formed on ISCU, but no Mössbauer analysis of iron bound to ISCU has ever been reported. Published EXAFS studies indeed showed that, counterintuitively, iron is bound to O/N ligands but not to sulfur, and therefore that iron is not bound to the cysteine-containing assembly site. In contrast, we show here that upon zinc removal, iron now binds to sulfur ligands. Further, we have identified the conserved amino acid of the assembly site as being the ligands of the iron ions. These findings have never been reported so far.

While the authors dismiss the role of frataxin in iron delivery, there is still a need to deliver iron, because free iron will not be found in a cell. While frataxin has been shown to serve a number of other roles (the specifics depending on the organism), nevertheless it is required to complex with ISCU and other ancillary proteins, most likely modulates the activities of other partners, but is still capable of delivering iron. If not frataxin, then how is iron delivered in the scheme presented in Fig. 6 of the manuscript?

Answer: In this work, we document by several means that Fe-S clusters can be assembled by reconstituted machineries according to two distinct paths, depending on presence or absence of zinc in the ISCU assembly site. We then conclude that Fe-S cluster assembly by Fe-ISCU constitutes the physiological process, but not the assembly by Zn-ISCU. Based on these results, we have been able to assign a biochemical function to FXN, which we think is physiologically relevant. FXN appears to stimulate Fe-S cluster assembly by accelerating persulfide transfer but is not required for iron insertion and FXN does not bind iron. Therefore, our data dismiss a role of FXN in iron delivery, which indicates also that another protein is required to provide iron. We hope to identify this protein in the future, but this is out of the scope of the present work.

The manuscript spends quite a bit of time discussing sulfur delivery, but how we go from no sulfide to two sulfide is not clearly defined by the experimental work described in the text and in Fig. 6. The

proposed ISCU dimer formation as a vehicle for bringing together two “Fe-S” pieces is rather speculative and not well supported by experimental data.

Answer: As pointed out by the reviewer, our data led us to propose the dimerization hypothesis as the only way to model our results. This hypothesis is indeed supported by the structure of the NFS1-ISC11-ACP complex reported by Cory et al. (PNAS 2017), in which the ISCU proteins are predicted to be very close to each other, thus opening the possibility that ISCU dimerizes in the complex. We hope to address this fascinating question in the future.

Overall, the reported work provides incremental advances, perhaps helps to clarify some prior published work, but certainly does not reveal the roles of Fd2 and frataxin in persulfide processing and does not provide a clearer definition of cluster assembly mechanism.

Answer: We hope that the reviewer will be more convinced by the revised version and our answers which provide here important findings and novel information regarding the mechanism of FXN- and FDX2-dependent Fe-S cluster assembly.

REVIEWERS' COMMENTS:

Reviewer #1 (Remarks to the Author):

In the revised version of the manuscript the authors have addressed most of the concerns raised during the initial submission. Based on the information provided I have few final recommendations to improve the clarity of the information presented before its publication.

1) Based on the author's response and the information provided in the document, I suggest the document to be revise in regards what constitutes sulfur transfer and persulfide reduction.

The experiments probing "sulfur transfer" events were initiated by the addition of cysteine substrate. Thus, these experiments probed the rate of persulfide formation and transfer combined, not the single step involving the persulfide transfer from NIA to U. Therefore, I recommend the authors to modify the text throughout the manuscript (abstract, main text, results, fig.4 and 6, and discussion) to indicate that kinetic experiments (Figure 4) determined rates of persulfide formation and transfer (gray bars).

Based on this experimental design, the rate of persulfide formation and transfer also depends on the concentration of Nsf1 and Cys present on these reactions. Since, these experiments were performed with stoichiometric amounts of cysteine, it is possible that under these experimental conditions the rate of the overall reaction is limited by the rate of persulfide formation, which may not be the limiting factor at conc. of cys above K_m .

The second set of experiments defined as "persulfide reduction" probed the rate of cluster formation monitored through UV-visible absorbance. That is, the rate defined as persulfide reduction is actually the rate of persulfide reduction and cluster formation combined (green bars). Whereas, the cluster assembly rate (orange bars) is the rate of the overall process. My recommendation is to make this point clear in the figure legend and in the text.

Reviewer #2 (Remarks to the Author):

The authors have responded by clarifying their presentation and improving the paper on many levels. Their reconstitution progress can lead to many more insights.

In the revised version of the manuscript the authors have addressed most of the concerns raised during the initial submission. Based on the information provided I have few final recommendations to improve the clarity of the information presented before its publication.

1) Based on the author's response and the information provided in the document, I suggest the document to be revised in regards what constitutes sulfur transfer and persulfide reduction.

The experiments probing "sulfur transfer" events were initiated by the addition of cysteine substrate. Thus, these experiments probed the rate of persulfide formation and transfer combined, not the single step involving the persulfide transfer from NIA to U. Therefore, I recommend the authors to modify the text throughout the manuscript (abstract, main text, results, fig.4 and 6, and discussion) to indicate that kinetic experiments (Figure 4) determined rates of persulfide formation and transfer (gray bars). Based on this experimental design, the rate of persulfide formation and transfer also depends on the concentration of Nsf1 and Cys present on these reactions.

Answer: We agree with the reviewer that our experimental setup for persulfide transfer probes a global reaction that combines persulfide formation and persulfide transfer to ISCU. However, as previously reported by us (Parent et al. Nat Commun 2015), persulfide formation by NFS1 is much faster (reaction completed in 10 s) than persulfide transfer itself (reaction completed in about 3 min with FXN and 30 min without) which was assessed using a persulfidated form of NFS1 devoid of L-cysteine and this rate was similar to the rate of the combined reaction (Supplementary Fig. 6a,b in Parent et al.). This indicates that persulfide transfer is the rate-limiting step of these two reactions, which implies that the observed rate of persulfide formation on ISCU upon addition of cysteine, is the rate of persulfide transfer.

Action: We have modified the text and the figure legend in the result section to clarify this point.

Since, these experiments were performed with stoichiometric amounts of cysteine, it is possible that under these experimental conditions the rate of the overall reaction is limited by the rate of persulfide formation, which may not be the limiting factor at conc. of cys above K_m .

Answer: Persulfide transfer is an internal reaction within the NIAU complex, after L-cysteine has reacted with NFS1, and therefore is not dependent on the concentration of L-cysteine. Nevertheless, it is conceivable that the reaction becomes limited by persulfide formation on NFS1 for concentration of L-cysteine below the K_m of NFS1 and thereby the rate of the reaction would depend on the concentration of cysteine. However, the fact that the rate of Fe-S cluster assembly is not changed from sub-stoichiometric to excess amounts of L-cysteine relative to the NIAU complex (see Supplementary Fig. 2b of the present paper, linear portion of the curves are superimposable), which indicates that cysteine is not rate-limiting at any of these concentrations, hence that persulfide transfer remains the rate limiting step.

The second set of experiments defined as "persulfide reduction" probed the rate of cluster formation monitored through UV-visible absorbance. That is, the rate defined as persulfide reduction is actually the rate of persulfide reduction and cluster formation combined (green bars). Whereas, the cluster assembly rate (orange bars) is the rate of the overall process. My recommendation is to make this point clear in the figure legend and in the text.

Answer: we agree that persulfide reduction also includes the rate of formation of the [2Fe2S] cluster.

Action: we have clarified this point in the text and the figure legends accordingly.